# Combining Gaussian Process with Hybrid Optimal Feature Decision in Cuffless Blood Pressure Estimation

**DOI:** 10.3390/diagnostics13040736

**Published:** 2023-02-15

**Authors:** Soojeong Lee, Gyanendra Prasad Joshi, Chang-Hwan Son, Gangseong Lee

**Affiliations:** 1Department of Computer Engineering, Sejong University, 209 Neungdong-ro, Gwangjin-gu, Seoul 05006, Republic of Korea; 2Department of Software Science & Engineering, Kunsan National University, 558 Daehak-ro, Gunsan-si 54150, Republic of Korea; 3Ingenium College, Kwangwoon University, 20 Kwangwoon-ro, Nowon-gu, Seoul 01897, Republic of Korea

**Keywords:** cuffless blood pressure estimation, Gaussian processing, optimal hybrid feature decision, F-test, robust neighbor component analysis, photolethysmography

## Abstract

Noninvasive blood pressure estimation is crucial for cardiovascular and hypertension patients. Cuffless-based blood pressure estimation has received much attention recently for continuous blood pressure monitoring. This paper proposes a new methodology that combines the Gaussian process with hybrid optimal feature decision (HOFD) in cuffless blood pressure estimation. First, we can choose one of the feature selection methods: robust neighbor component analysis (RNCA), minimum redundancy, maximum relevance (MRMR), and F-test, based on the proposed hybrid optimal feature decision. After that, a filter-based RNCA algorithm uses the training dataset to obtain weighted functions by minimizing the loss function. Next, we combine the Gaussian process (GP) algorithm as the evaluation criteria, which is used to determine the best feature subset. Hence, combining GP with HOFD leads to an effective feature selection process. The proposed combining Gaussian process with the RNCA algorithm shows that the root mean square errors (RMSEs) for the SBP (10.75 mmHg) and DBP (8.02 mmHg) are lower than those of the conventional algorithms. The experimental results represent that the proposed algorithm is very effective.

## 1. Introduction

The World Health Organization (WHO) announced that hypertension affects one billion people worldwide and causes 9 million deaths each year [1]. It is widely known that high blood pressure is the primary cause of death from cardiovascular disease (CVD) [1,2]. Hence, the accurate blood pressure (BP) measurement can have important public health implications. Recently, BP monitoring has been vital for people with CVD, especially the elderly living alone. Rapid changes in blood pressure in these people can mean that they have a severe illness because it is constantly changing due to intrinsic physiological changes for many causes, such as food, outside temperature, exercise, disease, and stress. Hence, precision, uncertainty, and accuracy in blood pressure measurements of physiological parameters have been a continuing concern for clinicians and practitioners [3,4,5]. Therefore, research on accurate BP measurement techniques is continuously needed.

Since about a decade ago, machine learning (ML) algorithms have been commonly used to estimate data in biomedical fields [4,6]. The ML algorithms, including support vector machine (SVM) [7], were utilized to estimate BP [8]. Wang et al. [9] introduced a method for BP estimation using a novel artificial neural network (ANN) and photoplethysmography (PPG) signals. Massaro et al. [10] proposed a decision support system for estimating health status based on artificial intelligence algorithms. A novel smart healthcare monitoring system using ML and the Internet of Things was developed by [11], which created an automated artifact detection method for BP and PPG signals. Tan et al. [12] introduced an artificial intelligence-enhanced BP monitoring wristband. The wristband’s sensors are based on piezoelectric nanogenerators. Nandi et al. [13] introduced a new long short-term memory and convolutional neural network using cuffless BP estimation based on the PPG and ECG signals. Recently, multi-channel PPGs were introduced using SVM ensemble-based continuous blood pressure estimation in [14]. Qiu et al. [8] proposed a new method for estimating BP using a window function-based piecewise neural network. This paper evaluated the random forest-based regression network and the three-layer ANN-based regression network as well as the SVM model as a less complex algorithm using the PPG signal in order to perform the accurate cuffless BP estimation. Here, valuable features were extracted using the PPG signal’s first and second derivative waveforms and used as input data for BP estimation. The critical issue to increasing ML algorithms’ reliability is extracting features essential to the response variable [15,16,17]. The two most popular methods for cuffless BP estimation are obtained using the extracted features and pulse transit time (PTT) from the PPG and electrocardiogram (ECG) signal pulses [8,18,19,20]. The feature extraction method using PTT effectively estimates BP because PTT is closely correlated with blood pressure [21]. Based on this principle, we can determine arterial pressure by measuring the pulse wave velocity (PWV) because PTT change corresponds to a change in PWV at a fixed distance, an indicator of BP variation [20,22].

Another issue for improving the performance of ML algorithms is feature selection to use as input data by replacing the original features [23,24,25,26,27]. Feature selection is an essential part of the learning algorithm’s performance, which selects a subset of features with high weights for the response variable, and eliminates duplicate features [15,16]. This process increases the reliability of ML, enhances predictive accuracy, and improves understanding. Irrelevant features provide no helpful information, and redundant features offer no more information than the currently selected feature [28]. In general, feature selection algorithms are classified into the filter, wrapper, and embedded algorithms.

First, the filter algorithm uses the available attributes of the training data independently of the learning algorithm [23,28]. Yang et al. [24] proposed neighbor component analysis (NCA) to learn feature weight vectors, which is efficient and computationally fast. However, the NCA algorithm may miss helpful features, so it can be combined with other methods to enhance performance. In addition, robust NCA (RNCA), which enhanced the performance of NCA, was introduced [29]. However, RNCA also has a problem: the number of features or weights may be specified and used as a fixed threshold heuristically when selecting the weight features.

Second, wrapper algorithms usually use learning algorithms to gain features and outperform filter algorithms in most cases. A wrapper algorithm demands one learning algorithm for feature selection and utilizes its performance to measure the superiority of a selected subset of features. However, wrapper algorithms are computationally intensive and sometimes difficult to handle in high-dimensional feature selection problems because the learning algorithm always needs to train each subset of features [24,28].

Third, the embedded algorithm is built into the learning algorithm. For example, the gradient descent algorithm is usually utilized to optimize the feature weights, indicating the relevance between the corresponding features and the target value. In particular, many embedded algorithms based on SVM have been introduced [24,25,26].

Various algorithms have been used to select valuable features [27]. Ding  et al. [30,31] proposed minimum overlap-maximum relatedness (MRMR) to obtain optimal subsets of multiple genes. Szabo et al. [32] introduced an algorithm that applies v-fold cross-validation combined with a randomly selected feature selection. Another algorithm that combines k-nearest neighbors with genetic algorithms was developed by Li et al. [33]. Guyon et al. [34] introduced an SVM-based approach called recursive feature elimination (RFE) to discover informative features.

In these regards, we propose a new methodology that combines the Gaussian process with hybrid optimal feature decision (CGHOFD) in cuffless blood pressure estimation. This study aims to accurately estimate systolic blood pressure (SBP) and diastolic blood pressure (DBP), which improves the reliability of cuffless BP estimation. Moreover, the Gaussian process (GP) algorithms are those that, like other kernel methods, can be precisely optimized for given hyperparameter values. Hence, it performs well due to well-optimized parameter values, especially on small datasets [35,36]. Another advantage of the GP is that it is robust to noisy signal and naturally regularizes [37]. Hence, the proposed CGHOFD algorithm uses a combined hybrid approach, such as minimum redundancy maximum relevance (MRMR) [30], ANOVA F-test (F-test) [38], and robust neighbor component analysis (RNCA) [29], to select weighted features among the original features. Although the MRMR, F-test, and RNCA algorithms can select features quickly, they also have the disadvantage of missing valuable features mentioned above. Therefore, the combined hybrid approach is to overcome this limitation. We then find the best feature set using the GP algorithm. The role of the GP as a learning algorithm is to determine feature subsets as the subset with the minimum root mean square error (RMSE). Here, we use a weighted feature subset as initial input through the feature selection method. The GP algorithm utilizes the input data to generate k-folds and perform cross-validation [29]. However, the GP algorithm uses more computer resources than other filter-based feature selection methods [24,28]. We intend to overcome the performance limitation of filter-based feature selection methods by combining the CGHOFD algorithm based on the biometric (PPG and ECG signals). We conduct extensive experiments to compare the conventional algorithms with the proposed CGHOFD algorithm on the public dataset for cuffless BPs estimation. The experimental results confirm that the proposed CGHOFD algorithm is very effective. To the authors’ knowledge, this is the first study of the proposed CGHOFD algorithm to estimate cuffless BPs. The CGHOFD algorithm is shown in the block diagram in Figure 1.

This paper is composed as follows. Section 2 contains the collection of PPG and ECG signals and preprocessing for feature extraction. The proposed combining Gaussian process with hybrid optimal feature decision (CGHOFD) algorithm is shown in Section 3. Section 4 denotes the experimental results and statistical analysis. Finally, the discussion and conclusion are denoted in Section 5 and Section 6.

## 2. Data Process

### 2.1. Data Set

We collected from the University of California Irvine (UCI) ML repository center [39], which was extracted from MIMIC-II (Multiple Parameter Intelligent Monitoring) data [39,40]. The database consists of ECG, finger PPG, and ABP (arterial blood pressure) signals from 3000 records (subjects) at 125 Hz (sampling frequency). Reference systolic blood pressure (SBP) and diastolic blood pressure (DBP) were calculated from ABP signals, and the feature set was obtained by combining PPG with ECG signal waveforms. Because each record range in the database was different, each record after 60 (s) was used to increase the reliability of the records obtained from the patients [41].

Hypertension ranges are classified into three conditions. BP between 140/80 and 159/99 mmHg is classified as stage 1 hypertension [42]. Stage 2 hypertension ranges from 160/100 to 179/109 mmHg [43]. Finally, BP above 180/120 mmHg is a hypertensive emergency, indicating very high BP leading to potentially life-threatening symptoms [44]. On the other hand, hypotension is BP less than 90/60 mmHg [45]. Therefore, it is vital for the body to adjust to rapid changes in BP and for patients to receive treatment. Additionally, the AAMI protocol recommends including blood pressure data of SBP > 160 mmHg 5% and DBP < 60 mmHg 5% [46]. In addition, based on the MIMIC II BP dataset of the recently published blood pressure estimation [47,48], we omitted records from specific subjects with very high and low BPs to remove abnormal outlier records, such as natural human physiological conditions as follows: (SBP >= 180, SBP <= 80, DBP >= 130, and DBP <= 50; unit:mmHg).

### 2.2. Preprocessing

We eliminated outliers using signal processings in order to extract useful features on the PPG and ECG signals. In the first, NaNs were eliminated across all signals to preserve alignment for each subject. The PPG signals were normalized in different values in each subject using the mini-max method. The ECG and ABP signals were not normalized to extract only time domain features, and preservation of the original ABP units (mmHg) was required to estimate SBP and DBP. We then used these to extract effective features from the PPG and ECG wave signals. In detail, we used a Kaiser window with a cutoff frequency of 35 Hz and a signal bandwidth of 3 dB to eliminate the noise of high-frequency. Next, the noise of low frequency was eliminated using a Kaiser window with a cutoff frequency of 0.0665 Hz and a bandwidth of 3 dB. After that, the ECG, PPG, and ABP signals were prepared into 20 (s). Segmented signals with minimum and maximum values above or below a certain threshold were discarded. The final step of preprocessing was to segment each set of 20 (s) windows of denoised ECG, PPG, and ABP signal into smaller segments containing fewer cardiac cycles, which provide input of the feature extraction. After the preprocessing, we acquired 1723 records.

### 2.3. Review of Feature Extraction

After preprocessing for accurate BP and CIs estimations, it is essential to extract valuable features using the PPG with ECG signals. Therefore, we analyzed the time and frequency domains of ECG and PPG signals. However, the frequency information was concentrated in the low-frequency band below 1.5 Hz, so valuable features could not be extracted in the frequency domain. Therefore, features were extracted using the pulse morphology of the PPG signal and the time between the ECG and PPG signals on the time axis as given in Figure 2. First, we obtained the pulse transit time (PTT), which is the time interval between the arrival of blood flowing distally and the opening of the aortic valve [8,39].

On the other hand, the pulse arrival time (PAT) denotes the time interval between the R peak of the ECG signal and the PPG rise points, and both PAT and PTT represent useful features for estimating BP values [8,18,19]. Another essential feature that is important to mention is the PPG’s pulse intensity ratio (PIR), which has also been represented to be inversely proportional to the diastolic trough [8]. We can observe the waveform associated with the heart rate cycle through the PPG signal. Hence, we define the PPG signal waves as a pulse, each corresponding to a cardiac cycle, with the rising edge as the systolic time (ST) and the pulse with the falling edge as the diastolic time (DT) [8]. In addition, the area in the pulse corresponding to ST is used as the systolic area (Sa), and the area in the pulse corresponding to DT is used as the diastolic area (Da). As shown in Figure 2, each pulse waveform is divided into these two areas. Therefore, we extracted features of each pulse, including Sa and Da, ST, DT, and cycle duration (CT), to extract features that effectively estimate BPs [8]. Figure 3 shows the first derivative of the PPGs and the second derivative of the PPGs. They are summarized in Table 1 and Table 2. Finally, we validate and evaluate the effectiveness of the final feature set using SVR, ANN, GP, and the proposed algorithms.

## 3. Combining Gaussian Process with Hybrid Optimal Feature Decision (CGHOFD)

### 3.1. Feature Weighting Using the F-Test

This paper uses the F-test as a hybrid mode to choose meaningful features for the BPs estimation [38]. The F-test is a statistical test that weights by computing the variance ratio. In this paper, the F-test based on one-way ANOVA calculates the between-group variance ratio and the within-group variance for each feature. In this case, a group represents instances with the same response value. Higher weights mean shorter intra-group distances and more considerable inter-group distances. Hence, features are ranked based on higher weights using the F-test based on one-way ANOVA. The null hypothesis is that the target values grouped by function in each F-test are drawn from populations with the same mean as opposed to the alternative hypothesis that the population means are not all equal:(1)$$H_{0}:\mu_{1}=\mu_{2}=\cdots =\mu_{m},\;H_{1}:\mu_{1}\neq \mu_{2}\neq \cdots \neq \mu_{m}$$
where *m* is the number of groups and $\mu_{m}$ denotes the mean for group *m*. The overall mean is calculated as
(2)$$\mu =\frac{1}{n}\sum_{k=1}^{m}\mu_{k}n_{k},\;\left(n=\sum_{k=1}^{m}n_{k}\right)$$
where $n_{k}$ is the number of the *k*th group. Hence, the mean of the real feature’s samples is given by
(3)$${\bar{x}}_{k}=\frac{1}{n}\sum_{i=1}^{n_{k}}x_{ik},\;$$The total mean is computed as
(4)$$\bar{x}=\frac{1}{n}\sum_{k=1}^{m}\sum_{i=1}^{n_{k}}x_{ik},\;$$
The sum of the mean squared deviation (MSD) within groups is given as
(5)$$S_{W}=\frac{1}{n}\sum_{k=1}^{m}\sum_{i=1}^{n_{k}}{\left(x_{ik}-{\bar{x}}_{k}\right)}^{2}$$
The sum of the MSD between groups is computed by
(6)$$S_{B}=\frac{1}{n}\sum_{k=1}^{m}\sum_{i=1}^{n_{k}}{\left({\bar{x}}_{k}-\bar{x}\right)}^{2}$$
Hence, we obtain the F-score as
(7)$$F_{S}=\frac{S_{B}/\left(m-1\right)}{S_{W}/\left(n-m\right)}$$
The F-test accepts the alternative hypothesis if *p* < 0.05. This means there is a difference in this feature between the two groups. If *p* ≥ 0.05, the null hypothesis is accepted, and the alternative hypothesis is rejected. This means there is no difference between the two algorithms when using this feature set. The smaller the *p*-value, the more significant the difference in this feature set between the two algorithms and, therefore, the more valuable it is for estimating BP. Therefore, a small *p*-value for the test statistic indicates the importance of a feature. Table 3 shows the ranked features obtained using the F-test.

### 3.2. NCA

Feature selection is choosing essential eatures from the original feature set. This means that only a few features really affect the target BPs. Hence, it is essential to reduce the dimension of the feature space while retaining only valid information for the BPs estimations. The weighted feature vectors are extracted from the original feature set using the NCA algorithm [29]. Here, the NCA method [24] trains a weighted feature vector by minimizing a loss function with diagonal adaptation that measures the mean deviation one-out regression loss from the training dataset. Hence, we defined a dataset as $T_{d}=\left\{\left(x_{i},y_{i}\right),i=1,\cdots ,n\right\}$. Here, we have chosen a weighted feature vector that utilizes the response vector *y* giving the explanatory vector *x* where $x\in R^{p\times n}$ and *n* are the number of observations. A regression was performed to randomly select the reference point γ(x) from $T_{d}$. Therefore, the response variable at *x* was included in the response variable at the reference point γ(x) [24].
(8)$$D_{w}=\sum_{m=1}^{p}w_{m}^{2}\left|x_{im}-x_{jm}\right|,$$
Here, $D_{w}$ is the weighted distance and $w_{m}$ denotes the weighted feature of *m*th. Thus, the probability $P(\gamma \left(x\right)=x_{j}|T_{d})$ that point *x* is chosen from $T_{d}$ as the reference point:(9)$$P\left(\gamma \left(x\right)=x_{j}|T_{d}\right)=\frac{k(D_{w}\left(x_{i}-x_{j}\right))}{\sum_{j=1}^{n}k\left(D_{w}\left(x_{i}-x_{j}\right)\right)}$$
Here, (k(z)=exp(−z/σ)) denotes the kernel, and the kernel width σ is a parameter that affects the probability that each point is chosen as a reference [24]. We assume that $P\left(\gamma \left(x\right)=x_{j}|T_{d}\right)\propto k\left(D_{w}\left(x_{i},x_{j}\right)\right)$ and estimate the response to $x_{i}$ using the training dataset in $T_{d}^{-i}$, $\left(x_{i},y_{i}\right)$. The probability that $x_{j}$ is chosen as the reference point for $x_{i}$ is given as
(10)$$\gamma_{ij}=P\left(\gamma \left(x\right)=x_{j}|T_{d}^{-i}\right)=\frac{k(D_{w}\left(x_{i}-x_{j}\right))}{\sum_{j=1,j\neq i}^{n}k\left(D_{w}\left(x_{i}-x_{j}\right)\right)}$$
(11)$$L_{i}=E\left(L\left(y_{i},{\hat{y}}_{i}\right)|T_{d}^{-i}\right)=\sum_{j=1,j\neq i}^{n}\gamma_{ij}L\left(y_{i},y_{j}\right)$$
where L is the loss function that gives the difference between $({\hat{y}}_{i}$, $y_{i})$. Thus, we apply the regularization parameter λ to minimize the loss function as follows:(12)$$F_{w}=\frac{1}{n}\sum_{i=1}^{n}L_{i}+\lambda \sum_{m=1}^{p}w_{m}^{2}$$
Hence, we use the regularization parameter to choose weighted feature vectors from high-dimensional features employing the NCA algorithm as λ(=0.015), [24] as given (Equation 8)–(Equation 12).

### 3.3. RNCA

The RNCA algorithm performance is affected by regularization parameters λ. Therefore, we need to define the RNCA algorithm to set the parameters effectively. Here, the regularization parameter is adapted utilizing the mean squared error and 5-fold cross-validation, as presented in (Algorithm 1: RNCA). Here, we applied a user-defined robust loss function given as $\zeta =1-\exp(-|y_{i}-y_{j}|)$. Hence, we decided the value of λ representing the minimum average loss value. Finally, we obtained the weighted feature vectors using the RNCA without selecting any other features, as shown in Table 3. Therefore, feature selection helps decrease the dimensionality to train the algorithm. However, as in Algorithm 1 (line: 9), when selecting RNCA weight features, it is problematic to designate the weight of features as a fixed threshold heuristically.

### 3.4. Minimum Redundancy Maximum Relevance (MRMR)

We can find the best subset of features that are maximally dissimilar to each other and can effectively represent the target variable using the MRMR algorithm [30]. The nature of the algorithm minimizes the redundancy of the feature set and maximizes the relevance of the feature set to the target variable. The aim of the MRMR method is to find an optimal subset *S* of features that maximizes $MA_{S}$, the relevance of *S* in terms of a target value *y*, and minimizes $MI_{S}$, the redundancy of *S*, where $MA_{S}$ and $MI_{S}$ are expressed with mutual information *I* as follows:(13)$$MA_{S}=\frac{1}{\left|S\right|}\sum_{x\in S}I\left(x,y\right),\;MI_{S}=\frac{1}{{\left|S\right|}^{2}}\sum_{x,z\in S}I\left(x,z\right)$$
where |S| denotes the number of features in *S*. We need to consider all $2^{\left|\phi \right|}$ combinations, where ϕ denotes the original feature set. Instead, the MRMR algorithm applies an additional forward process to provide feature rankings, which needs O(|ϕ|·|S|) computations using the mutual information quotient (MIQ) value.
(14)$${MIQ}_{x}=\frac{MA_{x}}{MI_{x}}$$
Here, $MA_{x}$ and $MI_{x}$ denote the relevance and redundacy of a feature, respectively, as 
(15)$$MA_{x}=I\left(x,y\right),\;MI_{x}=\frac{1}{\left|S\right|}\sum_{z\in S}I\left(x,z\right)$$
The MRMR algorithm ranks all features in ϕ and returns feature indices ordered by feature importance. So, the computational cost is $O(|\phi {|}^{2})$. This function uses a heuristic algorithm to quantify the importance of a feature and returns its weight. A large weight value indicates that the feature is required. In addition, decreasing the feature importance weights indicates confidence in feature selection. Therefore, we can use the output to find the optimal set S for a given number of features as given in Aigorithm 2.
(16)$$\max_{x\in S^{c}}{MIQ}_{x}=\max_{x\in S^{c}}\frac{I(x,y)}{\sum_{z\in S}I\left(x,z\right)}$$

**Algorithm 1** F-TEST and RNCA.
**Procedure** F-TEST(X, Y): training dataset01: return $\left(w_{f}\right)$ that produces weighted feature vectors using F-test02: select $(w_{f})\geq$ threshold
**End procedure**
 **Procedure** RNCA (X, Y): a training dataset01: partition training dataset into 5 folds**for** i=1,n **do**: where *n* is the number of the $\lambda_{i,k}$ line space02: $\lambda_{i,k}$: tuning using 5-fold cross-validation   **for** k=1,5 **do**:03:    call NCA(X, Y, $\lambda_{i,k}$): train NCA for λ regularization parameter04:    compute $L_{i,k}$: record loss values   **endfor**
**endfor**
05: $L_{\mu }$ = mean($L_{i,k}$): compute average loss value06: $\lambda_{b}=\arg\min_{L_{\mu }}\left(y|x,\lambda_{i,k},L_{\mu }\right)$: find best $\lambda_{b}$07: call NCA(X, Y, $\lambda_{b}$, ζ): ζ = $@\left(y_{i},y_{j}\right)1-\exp(-|y_{i}-y_{j}\left|\right)$08: return (w) that produces weighted feature vectors09: select (w)≥ (threshold) = 3; fixed threshold
**End procedure**



**Algorithm 2** MRMR algorithm.
**Procedure** MRMR(X, Y): training dataset01:   select ($\max_{x\in \phi }MA_{x}$): the most relevance feature02:   include (x⊂S): the selected feature *x* to an empty set *S*do:03:   find ($MA_{x}\neq 0,\in S^{c},MI_{x}=0,\in S^{c}$): where $S^{c}$ is the complement of *S*      if ($MA_{x}\neq 0,\notin S^{c}$) and ($MI_{x}=0,\notin S^{c}$) go to line 6:      else04:      select ($\max_{x\in S^{c},MI_{x}=0}MA_{x}$): the most relevance feature05:      include (x⊂S): the selected feature *x* to the set *S*      endifwhile: until $MI_{x}\neq 0$ for ∀ feature $\in S^{c}$do:06:   select ($\max_{x\in S^{C}}MIQ_{x}$): the feature with the nonzero relevance and redundancy07:   include (x⊂S): the selected feature *x* to the set *S*while: until $MA_{x}=0$ for ∀ feature $\in S^{c}$08:   include ($MA_{x}=0$): the feature to the set *S* in random order (x⊂S)
**End procedure**



### 3.5. The Proposed Hybrid Optimal Feature Decision (HOFD)

This paper proposes combining a Gaussian process with a hybrid optimal feature decision (CGHOFD) algorithm to determine valuable features for accurate blood pressure estimation. Feature selection generally consists of two stages; The first is to calculate the weight for each feature, and the second is to select the optimal subset to use as the input set. Hence, we choose one of the feature selection methods: RNCA [29], MRMR [30], and F-test [38] based on the proposed hybrid optimal feature decision (HOFD), as shown in Figure 4. Here, we can select RNCA via the hybrid mode (0) as shown in Figure 4 and steps 01 to 04 in Algorithm 3. Mode 0 is a filter-based RNCA algorithm [24,29], which uses the training dataset to obtain weighted features by minimizing the loss function. As mentioned in the section of RNCA, the best regularization parameter λ is obtained using steps 05 to 10 in Algorithm 3. The weighted feature vectors are then prepared using the RNCA, as shown in steps 11 to 12 of Algorithm 3. Afterward, we initialize variables to call the HOFD function shown in steps 13 to 16. Specifically, the weighted features were descending and sorted as in step 13. Next, the HOFD is called to compute the least root mean square error (RMSE) in step 17, where *X* denotes the training features and *y* is the reference of SBP. As shown in steps 18 to 20, we can obtain the least RMSE and the number of weighted features. To find the least RMSE, we continue to call the HOFD function by decreasing the number of weighted features by one in steps 17 to 21. Finally, the optimal feature index is determined to find the least RMSE in steps 22 to 25.
**Algorithm 3** main: GP-Based Hybrid Optimal Feature Decision (HOFD).01: hybrid = 0;**swich**(hybrid)   case 002:    [score,idx] = RNCA(X, Y)   case 103:    [score,idx] = F-test(X, Y)   case 204:    [score,idx] = MRMR(X, Y)**end****Procedure**: RNCA(X, Y): a training dataset05: partition training dataset into 5 folds**for** i=1,n **do**: where *n* is the number of the λ line space06:    $\lambda_{i,k}$: tuning using 5-fold cross-validation   **for** k=1,5 **do**:07:    call NCA(X, Y, $\lambda_{i,k}$): train NCA for λ regularization parameter08:    compute $L_{i,k}$: record loss values   **endfor****endfor**09: $L_{\mu }$ = mean($L_{i,k}$): compute average lossvalue10: $\lambda_{b}=\arg\min_{L_{\mu }}\left(y|x,\lambda_{i,k},L_{\mu }\right)$: find best $\lambda_{b}$11: call NCA(X, Y, $\lambda_{b}$, ζ): ζ = $@\left(y_{i},y_{j}\right)1-\exp(-|y_{i}-y_{j}\left|\right)$12: return (w) that produces weighted feature vectors13: [weights, indices] = sort(w, ’decent’): starting from weighted feature set14: num = 25; where num is number of features15: rmse = zeros(1, num);16: brmse=zeros(num,2);**for** j=1,num−1 **do**:17:    call HOFD( X( :, indices(1:num), y): 18:    return (ν): where ν is least rmse19:    brmse(j,1)=
ν;20:    brmse(j,2)= num;21:    num = num − 1;**endfor**22: [index] = min(brmse(:,1)):23: num = 25;24: num = num − index + 1;25: decision (indices(1:num)): the best feature subset**End procedure**

The proposed HOFD function starts from a set of weighted features obtained from RNCA and sequentially includes each feature that still needs to be selected to create a subset of candidate features. The HOFD conducts 5-fold cross-validation by repeatedly calling GP with different training subsets of X and y, as in step 03 of Algorithm 4. Here, Xtrain and ytrain contain identical X and y, while Xtest and ytest include the complementary subset of rows. Xtrain and Xtest have the features acquired from the columns of X that correspond to the current candidate feature set as in steps 04 to 07. Afterward, each time it is called, the GP should return a model criterion as in steps 08 to 09. Typically, GP uses Xtrain and ytrain to train; then, it predicts values for Xtest using that GP model in steps 10 and finally returns some measure of loss RMSE of those predicted values from the ytest in steps 11 to 12. After computing the mean criterion values for each candidate feature subset, HOFD chooses the candidate feature subset that minimizes the RMSE criterion value in step 13 of Algorithm 4, which is used to determine the best feature subset as given in (Algorithm 3). Hence, combining GP with HOFD leads to an effective feature selection process. In the first process of our algorithm, the filter-based algorithm is applied to find a weighted feature set. In the second process, GP directly determines the best feature subset from the weighted feature set. Therefore, we obtain the best feature subset with high weights from the original feature set as given in (Algorithms 3 and 4).
**Algorithm 4** function: HOFD.**function** [ν] = HOFD(X, y)01: m = 5; where m is the number of fold for cross-validation02: rmse = zero(1, m)03: cv = cvpartition (length (y), ’kfold’, m): cross-validation**for** k=1,m **do**:04:    Xtrain = X(cv.train(k),:)05:    ytrain = y(cv.train(k),:)06:    Xtest = X(cv.test(k),:)07:    ytest = y(cv.test(k),:)08:    call GP(Xtrain, ytrain): where GP denote the Gaussian process regression09:    return(mdl): return GP model parameters;10:    predict(’mdl’, Xtest): test GP model11:    return(ysp): estimated SBP values12:    $\mathrm{rmse}\left(k\right)=\mathrm{sqrt}(\mathrm{mean}\left({\left(\mathrm{ysp}-\mathrm{ytest}\right)}^{2}\right))$: evaluation criteria**endfor**13: ν = mean(rmse): where ν denotes least rmse**end**

### 3.6. GP Based on Bayesian Inference

This section describes GP regression [35], which is used to train and test the proposed HFD algorithm. Due to the size of the paper, the description of the conventional algorithms is omitted. The GP algorithm is a robust, flexible, and nonparametric Bayesian algorithm used in supervised ML [35]. In order to train the GP algorithm, the explanatory and response variables should be prepared as input and output data, respectively, as $D={\left\{x_{i},y_{i}\right\}}_{i=1}^{I}$, $x\in R^{I\times D}$, and $y\in R^{I\times 1}$. Here, we used a mapping function $f_{m}=f\left(x\right)$ to estimate *y* given *x*. Hence, we assumed that the response variable *y* is acquired using the corresponding $x^{T}w$ by including noise, as follows
(17)$$y=x^{T}w+\varepsilon ,\;\varepsilon ∽N\left(0,\sigma^{2}I\right)$$
The weighted vectors *w* and variance $\sigma^{2}$ were acquired from the resampled signal dataset. The GP algorithm estimates the response variable based on Gaussian processes (GP) using the mapping function $f_{m}\left(x\right)$ and explicit essential functions β.
(18)$$f_{m}\left(x\right)∽\mathrm{GP}\left(0,k\left(x,x^{{}^{\prime }}\right)\right)$$
where $f_{m}\left(x\right)$ is acquired from a zero-mean GP algorithm using a covariance function $k(x,x^{{}^{\prime }})$ [36]. Hence, we can obtain the mapping function $f_{m}\left(x\right)=\beta {\left(x\right)}^{T}w$. The mean function of the input data can be defined as the expected value of the mapping function $\theta \left(x\right)=E[f_{m}\left(x\right)]$. A latent variable covariance function obtains the smoothness of the response variables, and the basic function projects the input data *x* into the dimensional feature space.
(19)$$k\left(x,x^{{}^{\prime }}\right)=E\left[\left(f_{m}\left(x\right)-\theta \left(x\right)\right){\left(f_{m}\left(x^{{}^{\prime }}\right)-\theta \left(x^{{}^{\prime }}\right)\right)}^{T}\right]$$
We define the expected value of (Equation 19) as
(20)$$k\left(x,x^{{}^{\prime }}|\eta \right)\approx \sigma^{2}\exp\left(-\frac{{∥x-x^{{}^{\prime }}∥}^{2}}{2\eta^{2}}\right)$$
where k is a kernel for the GP [35], η is a hyperparameter, and $\sigma^{2}$ is a variance based on resampled signals. In the study, we use exponential squares as the kernel, as in (Equation 20). Thus, the kernel decides the properties of the mapping function $f_{m}\left(x\right)$. We can define an instance of response variables y using the Bayesian inference based GP as
(21)$$p\left(y_{i}|f_{m}\left(x_{i}\right),x_{i}\right)∽N\left(y_{i}|\beta {\left(x_{i}\right)}^{T}w+f_{m}\left(x_{i}\right),\sigma^{2}\right)$$
where $\beta (x_{i})$ denotes a basic function transforming the original explanatory variable *x* into a new variable β(x). Thus, we determined $\Theta =\{w,\eta ,\sigma^{2}\}$ from the dataset D, and the marginal likelihood is expressed as
(22)$$p\left(y|x\right)=p\left(y|x,\Theta \right)\approx N(y|\Omega w,k\left(x,x^{{}^{\prime }}|\eta \right)+\sigma^{2}I),$$
Generally, the local maxima for the hyperparameter Θ can be determined and used to train the GP algorithm. In addition, choosing an appropriate kernel depends on hypotheses, such as the smoothness and expected patterns of the data. By maximizing the log marginal likelihood, we can estimate the hyperparameter Θ, as follows
(23)$$\begin{array}{cc} \log p\left(y|x,\Theta \right)= & -\frac{1}{2}\log\left|k\left(x,x^{{}^{\prime }}|\eta \right)+\sigma^{2}I\right|-\frac{1}{2}i\log2\pi \\ & -\frac{1}{2}{\left(y-\Omega w\right)}^{T}{\left[k\left(x,x^{{}^{\prime }}|\eta \right)+\sigma^{2}I\right]}^{-1}\left(y-\Omega w\right) \end{array}$$
where $k(x,x^{{}^{\prime }}|\eta )$ is the kernel matrix and Ω denotes the matrix of the explicit basic function. Herein, we apply a penalty-fitting scale to represent the logarithmic likelihood and maximize it using a gradient approach using optimization techniques. The hyperparameters $\Theta =\{w,\eta ,\sigma^{2}\}$ using the GP algorithm maximize the likelihood p(y|x) as a function of Θ.
(24)$$L\left(\hat{\Theta }\right)=\arg\max_{\Theta }\log\left(y|x,\Theta \right)$$
First, we determine $\hat{w}\left(\eta ,\sigma^{2}\right)$ to predict hyperparameters that maximize the log-likelihood concerning *w* for a given $\left(\eta ,\sigma^{2}\right)$ as
(25)$$\begin{array}{c} \hat{w}\left(\eta ,\sigma^{2}\right)={\left\{\Omega^{T}{\left[k\left(x,x^{{}^{\prime }}|\eta \right)+\sigma^{2}I\right]}^{-1}\Omega \right\}}^{-1}\Omega^{T}{\left[k\left(x,x^{{}^{\prime }}|\eta \right)+\sigma^{2}I\right]}^{-1}y \end{array}$$

Second, we need a probability density function $p(y^{*}|y,x,x^{*})$ for the probabilistic estimation of the Bayesian GP algorithm using known hyperparameters. However, we estimated a response variable *y* using a finite amount of new input data $x^{*}$ and predicted the output of these data based on a multivariate Gaussian distribution with a kernel-generated covariance matrix. Thus, we denote the conditional probability distribution as follows.
(26)$$p\left(y^{*}|y,x,x^{*}\right)=\frac{p(y^{*},y|x,x^{*})}{p(y|x,x^{*})}$$
In order to acquire the joint density probability function in the numerator, as expressed in (Equation 26), the mapping functions $f_{m}^{*}$ and $f_{m}$ should be used as follows.
(27)$$\begin{array}{cc} p\left(y^{*},y|x,x^{*}\right) & =\int \int p\left(y^{*},y,f_{m}^{*},f_{m}|x,x^{*}\right)dfdf^{*}\\ & =\int \int p\left(y^{*},y|f_{m}^{*},f_{m},x,x^{*}\right)p\left(f_{m}^{*},f_{m}|x,x^{*}\right)dfdf^{*} \end{array}$$
The GP algorithm assumes that each response variable $y_{i}$ depends only on the corresponding latent variable $f_{m}\left(x_{i}\right)$ and input vector $x_{i}$. Given y,x and the hyperparameters Θ, the expected value of the estimation is given as:(28)$$\begin{array}{cc} E(y^{*}|y,x,x^{*},\Theta ) & =\theta {\left(x^{*}\right)}^{T}w+c\left(x,x^{{}^{\prime }}|\eta \right)\varphi \\ \; & =\beta {\left(x^{*}\right)}^{T}w+\sum_{i=1}^{I}\varphi_{i}k\left(x^{*},x_{i}|\eta \right) \end{array}$$
where $\varphi ={\left[k\left(x,x\right)+\sigma^{2}I\right]}^{-1}\left(y-\Omega w\right)$. Practically, we determined an optimal point prediction ${\hat{y}}^{*}$ based on the loss function as
(29)$$E_{L}\left({\hat{y}}^{*}|x^{*}\right)=\int L\left(y^{*},{\hat{y}}^{*}\right)p\left(y^{*}|x^{*},D\right)dy^{*}$$
We obtained and predicted $y^{*}\approx {\hat{y}}^{*}$ and minimized the expected value of the loss function $L(y^{*},{\hat{y}}^{*})$ by minimizing between $y^{*}$ and ${\hat{y}}^{*}$ as
(30)$${\hat{y}}_{\mathrm{opt}}|x^{*}=\arg\min_{{\hat{y}}^{*}}E_{L}\left({\hat{y}}^{*}|x^{*}\right)$$
In this study, we used the mean absolute error (MAE), RMSE, and mean error (ME) as loss functions, as given by: L.

## 4. Experimental Results

### 4.1. ML Model’s Complexity and Parameter Tuning

We adjusted the parameters for the proposed and conventional algorithms before the training step. These parameters are essential because they can improve algorithm performance when configured effectively. This study used 5-fold cross-validation to fine-tune parameters. First, we defined the core parameters for each algorithm and determined the range of possible values for each parameter. First, we performed the grid search on all the possible combinations of the parameters for each ML algorithm, finding the best parameter sets that assist the ML algorithms in obtaining the highest results. Next, we conducted a 5-fold cross-validation to improve the algorithm’s robustness with the optimal parameters obtained. Then, we randomly separated the training data into five non-overlapping subsets of equal size. There were five iterations, four folds were used for learning each iteration, and the remaining folds were applied to perform the evaluation. The final output was the average of the five folds. Table 4 shows the parameter ranges for each parameter of each ML algorithm under consideration and the optimized parameters after conducting the grid search. We also computed the feature training and testing times using MATLAB^®^ 2022 [49], as shown in Table 5. As a result, the proposed algorithm requires more computational times than the GP algorithm using the public dataset.

### 4.2. Datasets and Analysis

In this paper, the MIMIC II database was used to extract the PPG, ECG, and ABP signals [39,40]. The first dataset comprised 3000 records (subjects) with the PPG, ECG, and ABP signals. Because each record range in the database was different, each record after 60 (s) was used to increase the reliability of the records obtained from the patients [41]. After preprocessing, the final database was composed of 1723 records with unique subjects, which satisfied the general population 85 subjects [46]. Table 6 and Figure 5 represented some statistical information about the range and distribution of the reference SBP and DBP in the final dataset.

### 4.3. Evaluation Metrics

The feature set was randomly split into 80% for training and 20% as testing, respectively. Next, the reference BPs (systolic and diastolic) were obtained from the ABP envelope signals, as shown in Figure 2c. First, to evaluate the experimental results, we compared the proposed with the conventional methods using the mean error (ME) and standard deviation (SDE) of ME as shown in Table 7. The ME and SDE between the estimated BPs and the calculated reference BPs were calculated by the recommendations of the Association for the Advancement of Medical Instrumentation (AAMI) protocol [50]. A device can pass the AAMI protocol if its measurements’ error has an ME value of less than five mmHg with an SDE of less than eight mmHg [50]. As shown in Table 8, we used mean absolute error (MAE) results as an evaluation method for the proposed algorithm. Furthermore, based on the results of the MAE and SDE, we also calculated the probability of the British Hypertension Society (BHS) protocol [3], as shown in Table 9. The average error of the proposed algorithm was calculated by $er_{i}=(ep_{i}-rp_{i}$), for each record *i*, where ep denotes the estimated BPs (SBP or DBP) and rp is a reference BP. Hence, the mean error (ME) and MAE were specifically given as ($\frac{1}{n}\sum_{i=1}^{n}er_{i}$) and ($\frac{1}{n}\sum_{i=1}^{n}\left|er_{i}\right|$), respectively. All results were obtained as the average of 30 experiments for each algorithm.

The coefficient of determination (R-squared) is the original formula for quantifying the degree to which the independent variable determines the dependent variable in terms of the proportion of variance. For example, a value such as $R^{2}$ = 0.8 indicates good regression algorithm performance regardless of the range and distribution of ground truth values. On the other hand, the root mean square error (RMSE) and MAE values equal to 0.7 do not tell us anything about the quality of the regression performed. Therefore, the coefficient of determination is more informative and accurate than MAE and RMSE [51]. Therefore, we computed the coefficient of determination $R^{2}$ ranging from 0 to 1, with values closer to 1 indicating a stronger relationship between the predictor and response variables, as shown in Table 10.

Here, we again distinguished between GPs with RNCA and proposed CGPs with RNCA. GP with RNCA is an algorithm for determining a weighted feature vector through a conventional RNCA algorithm and using it as an input to a GP algorithm. On the other hand, the combining GP (CGP) and RNCA is an algorithm in which the proposed HOFD process finally determines the weighted feature vectors selected by the RNCA method as given in Algorithms 3 and 4. Please refer to Figure 4 and Algorithms 3 and 4.

Mean values acquired using different algorithms under different conditions are prevalent in biomedical processing [52]. Therefore, comparing two or more algorithms means that the *t*-test alternative one-way analysis of variance (ANOVA) is appropriate. Since ANOVA depends on the same stochastic distribution as the *t*-test, the interest of ANOVA lies in the location of the indicated distribution. Therefore, the ANOVA is used to evaluate the relative variance size of the mean variance between groups compared to the within-group mean variance [53], as shown in Figure 6.

## 5. Discussion

This is the first study to propose combining the Gaussian process with hybrid optimal feature decision (HOFD) in cuffless blood pressure estimation. Table 3 shows the high-scoring features selected using the F-test and RNCA algorithms [29,38]. Here, we can see that the ranked features rely on the feature selection algorithm. In addition, the ranked features were changed depending on the SBP or DBP target variable. Hence, we confirm the best feature decision using the proposed HOFD algorithm. Therefore, the proposed HOFD algorithm was more conducive to improving the performance of ML than using the threshold of the conventional RNCA algorithm. Table 5 confirms that the proposed algorithm is more complex than the GP algorithm in terms of computational complexity. This indicates that during the HOFD process, computing resources were consumed in deciding the weight feature subset according to evaluation criteria.

Based on the results of evaluations according to the AAMI/ESH/ISO protocols [46], we show the SDEs of MEs for the SBP (13.33 mmHg) and DBP (11.50 mmHg) acquired using the SVM algorithm compared with the reference BPs. The ANN algorithm represents the SDEs of MEs for the SBP (13.04 mmHg) and DBP (10.32 mmHg). The GP algorithm shows the SDEs of MEs for the SBP (11.80 mmHg) and DBP (9.61 mmHg). In addition, we show the SDEs of MEs for the SBP (10.85 mmHg) and DBP (8.42 mmHg) obtained using the GP with the RNCA algorithm. As given in Table 7, we observe the SDEs of MEs for the SBP (11.62 mmHg) and DBP (9.44 mmHg) obtained using the CGP with F-test algorithm, and we show the SDEs of MEs for the SBP (11.31 mmHg) and DBP (8.89 mmHg) acquired using the CGP with MRMR algorithm. Furthermore, the proposed CGPRNCA algorithm is obtained in the lower SDEs of MEs for the SBP (10.79 mmHg) and DBP (8.07 mmHg) compared with the conventional SVM, ANN, GP, and GP with RNCA algorithms. Although the proposed methodology does not meet the AAMI/ESH/ISO protocols [46] in the SBP results, it is superior to the conventional methods. Furthermore, it shows the possibility of future development, and the DBP results are very close to the protocol criteria. Figure 7a was well represented to compare the performance of the proposed CGPRNCA algorithm with the reference ABP (mmHg) concerning the SDE of ME for SBP. Figure 7b compares the performance of the proposed CGPRNCA algorithm with the reference ABP (mmHg) concerning the SDE of ME for DBP. The Bland–Altman plots (c) and (d) compare the performance of the SVM algorithm with the reference ABP (mmHg) concerning the SDEs of MEs for SBP and DBP, as represented in Figure 7.

The MAEs of SBP (9.84 mmHg) and DBP (8.30 mmHg) acquired using the SVM algorithm are compared to the reference BPs given in Table 8. The ANN algorithm shows the MAEs of SBP (9.75 mmHg) and DBP (7.23 mmHg) compared with reference BPs. The GP algorithm represents the MAEs of SBP (8.39 mmHg) and DBP (6.45 mmHg) compared with reference BPs. Table 8 also shows the MAEs of SBP (8.93 mmHg) and DBP (6.45 mmHg) acquired using the GP with the RNCA algorithm (GPRNCA). We show the MAEs of SBP (7.83 mmHg) and DBP (5.97 mmHg) obtained using the CGP with the F-test algorithm and observe the MAEs of SBP (7.85 mmHg) and DBP (5.82 mmHg) obtained using the CGP with MRMR algorithm. Regarding estimation accuracy, the proposed CGPRNCA algorithm exhibits the lowest MAE for SBP (7.22 mmHg) and DBP (5.18 mmHg) compared with the reference BPs given in Table 8. In particular, SBP 2.2% = (7.38−7.22)/7.22×100 and DBP 6.2% = (5.50−5.18)/5.53×100 were improved compared to the MAEs of SBP (7.38 mmHg) and DBP (5.50 mmHg) in the GPRNCA algorithm, proving that the proposed CGPRNCA algorithm is effective in estimating SBP and DBP. We confirm the effect of the HOFD process that finally determines the weighted feature subset. In addition, when the proposed CGPRNCA algorithm is compared with the GP algorithm, it shows an improved performance of 16.2% = (8.39−7.22)/7.22×100 for SBP and 24.5% = (6.45−5.18)/5.18×100 for DBP. On the other hand, the SDEs of MAEs in all algorithms show stable values except for the SVM, as given in Table 8. The comparison between the SBP estimation result of the proposed CGPRNCA algorithm and the estimation result of the conventional SVM is well confirmed in Figure 8.

Moreover, we compared the CGPRNCA algorithm with the SVM, ANN, GP, GPRNCA, CGPF-Test, and CGPMRMR algorithms following the British hypertension protocol (BHS) [3]. We evaluated the mean absolute error (MAE) for three groups of less than five mmHg, less than ten mmHg, and 15 less than mmHg, respectively. The readings using the proposed CGPRNCA algorithm were 54.58% (≤5 mmHg), 75.31% (≤10 mmHg), and 86.21% (≤15 mmHg) for the SBP in the test scenario, and 65.78 % (≤5 mmHg), 84.83% (≤10 mmHg), and 93.21% (≤15 mmHg) for the DBP in the test scenario. The probabilities of the proposed CGPRNCA algorithm based on the BHS are higher than those obtained by the conventional SVM, ANN, GP, and GPRNCA, CGPF-Test, and CGPMRMR algorithms, as given in Table 9. Compared to the BHS protocol, the proposed CGPRNCA algorithm obtained a grade of C in SBP and B in DBP [3]; it sets a standard for improving the performance of the future algorithm as shown in Table 9. The MAEs were 54.58% (≤5 mmHg), 75.31% (≤10 mmHg), and 86.21% (≤15 mmHg), respectively, for SBP and 65.78% (≤5 mmHg), 84.83% (≤10 mmHg), and 93.21% (≤15 mmHg), respectively, for DBP as expressed in Table 9. Furthermore, we observe that the proposed CGPRNCA algorithm indicates better accuracy than the conventional algorithms for the cuffless BP estimations.

As shown in Figure 6, we display the ANOVA experiments from RMSEs between the conventional and proposed algorithms for the SBP and DBP. Here, since the *F* values follow the *F* distribution, it can be concluded that the *F* values obtained from the observations can be compared with the threshold value α(=0.05) in the *F* table. The RMSEs of SVM, ANN, and GP are significantly different, as given in Figure 6a. The *p*-value of 0.0009 is lower than the critical value (0.05). Box (b) shows that the RMSE of the GP with RNCA (GPRNCA) is lower than that of the CGP with F-test (CGPF-TEST). There is a statistically significant difference between the GPRNCA and CGPF-TEST, according to comparisons *p*(=9.0 × 10${}^{-05}$). The results of the CGP with RNCA (CGPRNCA) show a statistically significant difference *p*(=0.0014) from that of the CGP with MRMR (CGPMRMR), according to comparison *p*(=0.05) value, as shown in Figure 6b. However, there is no statistically significant difference between the GPRNCA and CGPRNCA, according to comparisons *p*(=0.77), as shown in Figure 6b. The result of the GP is statistically different from that of the SVM and ANN, according to comparison *p*(=3.35 × 10${}^{-11}$), as given in Figure 6c. Figure 6d also shows the RMSEs for the GPRNCA and CGPF-TEST regarding the reference DBP values. Again, the GPRNCA is statistically different *p*(=9.4 × 10${}^{-09}$) from the CGPF-TEST, according to comparison *p*(=0.05). A statistically significant difference *p*(=1.7 × 10${}^{-06}$) in the RMSEs of the CGPMRMR and CGPRNCA is observed concerning the reference DBP values. A statistically significant difference *p*(=0.02) in the GPRNCA and the proposed CGPRNCA is observed concerning the reference DBP values, as shown in Figure 6d.

In addition, the $R^{2}$ values of the GP algorithm for SBP (0.80) and DBP (0.67) indicate a stronger relationship with the response than those of the ANN algorithm for SBP (0.74) and DBP (0.59), as given in Table 10. The $R^{2}$s of the GPRNCA algorithm for SBP (0.83) and DBP (0.75) show higher relationships for the response variable compared to the CGPF-Test algorithms, as given in Table 10. The $R^{2}$s of the CGPMRMR algorithm for SBP (0.82) and DBP (0.72) represent a lower relationship with the response variable than that of the CGPRNCA algorithm. The $R^{2}$s of the proposed CGPRNCA algorithm for the SBP (0.83) and DBP (0.77) also indicate a higher relationship with the response variable than that of the conventional SVM, ANN, GP, and CGPMRMR algorithms, as given in Table 10. Figure 9a shows the $R^{2}$ value between the proposed CGPRNCA and reference SBP (mmHg). The $R^{2}$ value between the proposed CGPRNCA and reference DBP (mmHg) is given in Figure 9b. Based on the results, the $R^{2}$ values of the CGPRNCA algorithm for SBP and DBP indicate a stronger relationship with the response than those of the SVM algorithm for SBP and DBP, as given in Figure 9.

The RMSEs of the GP algorithm for the SBP (11.75 mmHg) are lower than that of the SVM algorithm for the SBP (13.27 mmHg). The RMSEs of the GP algorithm for DBP (9.55 mmHg) are lower than that of the SVM algorithm for DBP (11.46 mmHg). However, the RMSEs of the GPRNCA algorithm for the SBP (10.79 mmHg) and DBP (8.38 mmHg) are slightly lower than those of the CGPF-Test algorithm for the SBP (11.59 mmHg) and DBP (9.41 mmHg), as given in Table 10. We also confirm the RMSEs of SBP (11.25 mmHg) and DBP (8.85 mmHg) obtained using the CGPMRMR algorithm. The RMSE results were compared with reference ABP using the proposed CGPRNCA algorithm for SBP (10.75 mmHg) and DBP (8.02 mmHg). These results represent 9.3% = (11.75−10.75)/10.75×100 and 19.1% = (9.55−8.02)/8.02×100 performance improvements for SBP (11.75 mmHg) and DBP (9.55 mmHg), respectively, compared to a conventional GP algorithm. The results confirm that the proposed CGPRNCA algorithm is more accurate than conventional algorithms for cuffless BPs estimation. In addition, the proposed methodology can constantly monitor BP change through the continuous variability of RMSE’s SDEs and MAE’s SDEs to estimate hypertension risk. The proposed HOFD process based on the GP algorithm may effectively be used for BP estimation.

### Limitation

Although we experimented with a public MIMIC-II dataset, this study showed limitations in that patient distribution and experimental results did not satisfy the AAMI/ESH/ ISO protocols [46]. Based on the AAMI/ESH/ISO protocol, we evaluated the validity of the proposed method. The MIMIC II satisfies the standard of 85 subjects in general, the MIMIC II dataset does not meet the AAMI/ESH/ISO protocol at SBP >160 mmHg 5%, >140 mmHg 20%, and DBP >100 mmHg 5%, >85 mmHg 20% [46]. The high blood pressure part shows a small distribution, and the low blood pressure part shows a large distribution as 17.9% for SBP 100 mmHg or less and 38.0% for DBP 60 mmHg or less. Moreover, obtaining inter- and intra-individual BP variations is crucial for cuffless device evaluation but challenging to acquire [54]. Finally, the subject population should exhibit a wide BP range for calibration-free algorithms, such as those required by universal standards. However, identifying these populations can be difficult and costly, and meeting many of these criteria will require much work, especially in lab-scale studies of popular cuffless devices [54]. We believed that MIMIC II data were obtained from long-term monitoring of patients. However, further research should be considered when obtaining data based on the AAMI/ESH/ISO protocol. Therefore, our laboratory will conduct a confidence interval of BP estimation to track the inter-and intra-individual BP variations to observe the cardiovascular functions’ variance over 24 h. In addition, we should obtain more BP data and conduct experiments to verify intra-individual blood pressure changes according to the AAMI/ESH/ISO protocol [54]. Moreover, since calibration-free algorithms often use age and gender as inputs to ML models, this causes the accuracy of the experimental results to be unclear [54]. However, in this study, age and gender were not used as input values for the proposed algorithm. Nevertheless, the following study should improve the performance of the CGPRNCA algorithm by reducing the SDE of ME to pass the AAMI criteria [50]. Another disadvantage is that the complexity of the HOFD algorithm should be improved for BPs estimation.

## 6. Conclusions

In conclusion, the proposed CGPRNCA improved accuracy and stability by using the HOFD process that reduces uncertainties such as MAE, SDE of mean error (ME), and RMSE for cuffless SBP and DBP estimation. The proposed CGHOFD algorithm selects a filter suitable for feature characteristics using a hybrid mode to improve the disadvantages of the conventional filter-based method. Then, it is a combining method in which the selected weighted subsets are finally determined by finding the best feature subsets using the GP algorithm evaluation criterion. We performed extensive experiments to compare the conventional algorithms with the proposed CGHOFD algorithm on the public dataset for cuffless BPs estimation. The experimental results confirm that the proposed CGHOFD algorithm is very effective. This study contributes to BPs estimation as follows. First, the proposed combining Gaussian process with a hybrid optimal feature decision (CGHOFD) algorithm is developed to improve reliability in cuffless BP estimation. Second, the proposed methodology using the hybrid optimal feature decision (HOFD) process overcomes the limitation of missing valuable features, which is a limitation of the conventional filter-based feature selection algorithm. Third, the adaptive HOFD algorithm automatically uses GP evaluation criteria to decide the best feature subset. Fourth, the proposed method solves the problem of specifying the number of weights when selecting the weighted features and the RNCA problem using a fixed threshold.

## Figures and Tables

**Figure 1 diagnostics-13-00736-f001:**
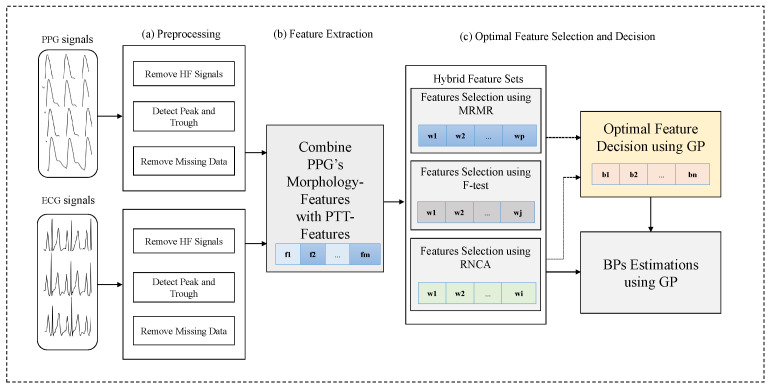
Block diagram of the proposed combining optimal hybrid feature selection and decision based on Gaussian process (GP) algorithm.

**Figure 2 diagnostics-13-00736-f002:**
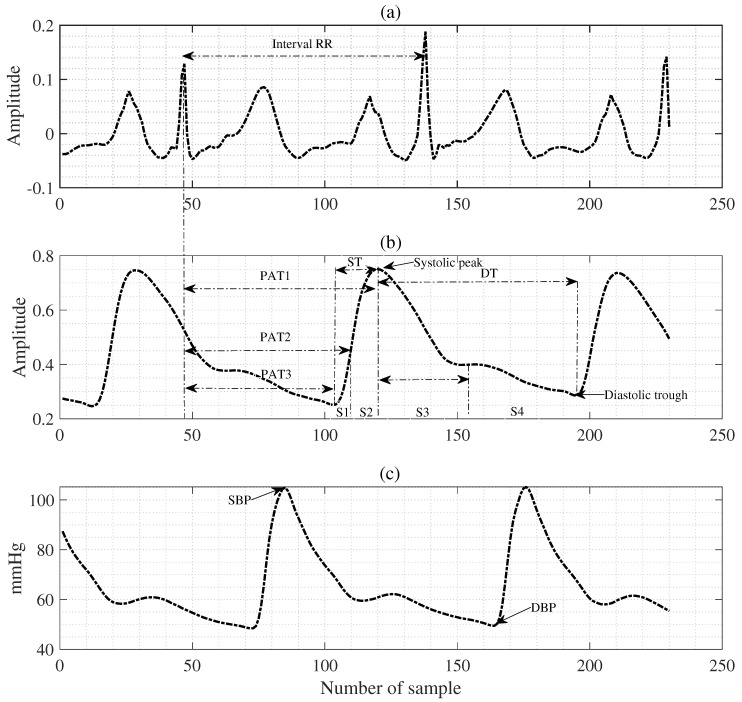
Features extraction (I) from the PPG with ECG signal, where (**a**) is an ECG signal example, (**b**) denotes a PPG signal example, and (**c**) is a target signal (ABP) example.

**Figure 3 diagnostics-13-00736-f003:**
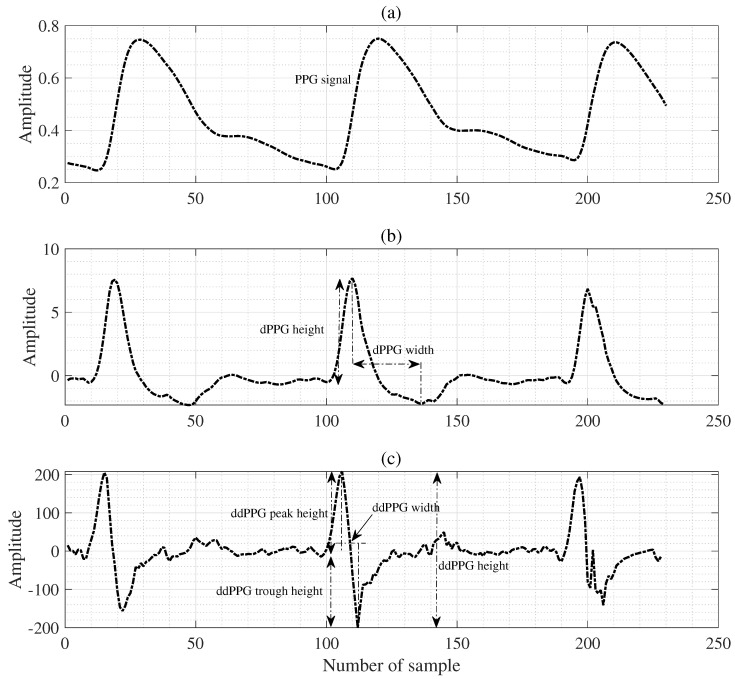
Features extraction (II) from the 1st and 2nd derivative of PPG waveforms, where (**a**) is a PPG signal example, (**b**) denotes the 1st derivative of PPG wave signal, and (**c**) denotes the 2nd derivative of PPG wave signal.

**Figure 4 diagnostics-13-00736-f004:**
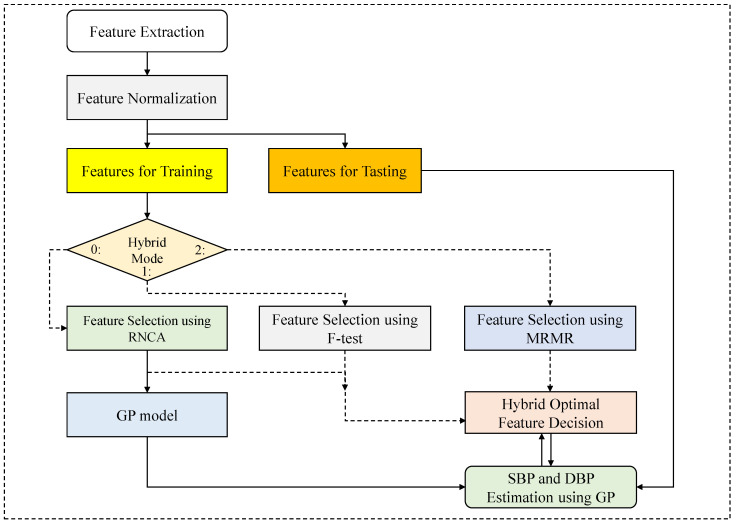
The proposed combining Gaussian process (GP) with hybrid optimal feature decision (HOFD) procedure.

**Figure 5 diagnostics-13-00736-f005:**
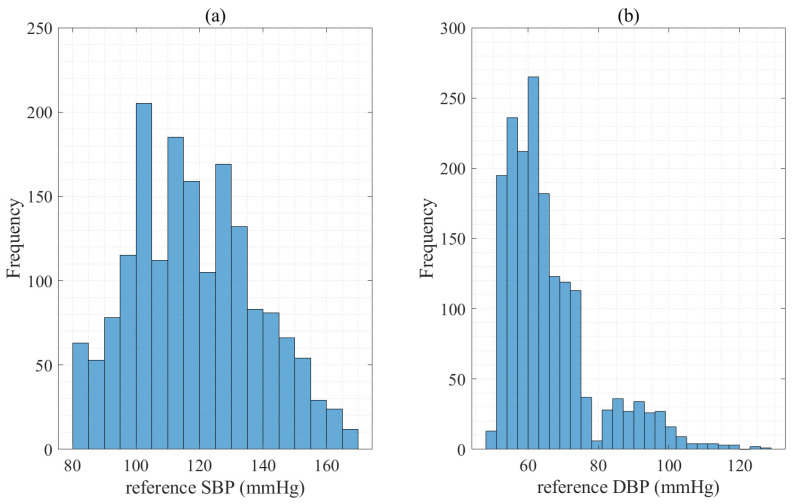
Histogram of the database, where (**a**) denotes the reference SBP (mmHg) and (**b**) denotes the reference DBP (mmHg).

**Figure 6 diagnostics-13-00736-f006:**
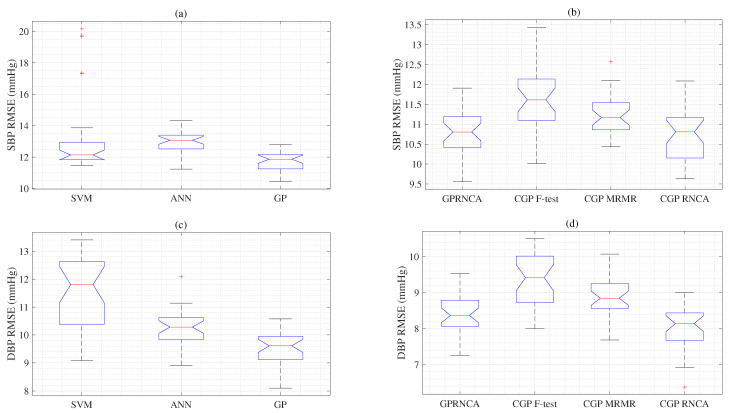
We compared the performance between the SVM, ANN and GP with respect to the reference ABP for the SBP (**a**) and DBP (**c**). We also compared the performance between the GPRNCA and CGP with an F-test concerning the reference ABP for the SBP (**b**) and DBP (**d**); We compared the performance between the CGP with MRMR and CGP with RNCA with respect to the reference ABP for the SBP (**b**) and DBP (**d**); Finally, we compared the performance between the GP with RNCA and CGP with RNCA with respect to the reference ABP for the SBP (**b**) and DBP (**d**).

**Figure 7 diagnostics-13-00736-f007:**
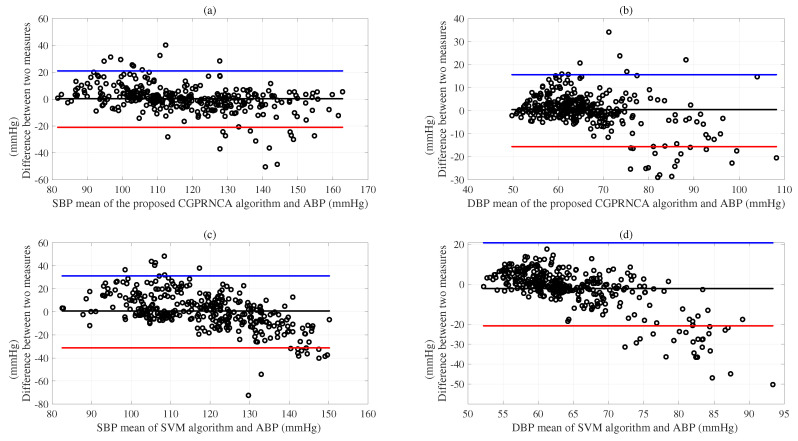
Top panel (**a**) shows a comparison of the performance between the proposed CGPRNCA and reference SBP (mmHg); panel (**b**) compares the performance between the proposed CGPRNCA and reference DBP (mmHg); panel (**c**) compares the performance between the SVM and reference SBP (mmHg); panel (**d**) compares the performance between the SVM and reference DBP (mmHg).

**Figure 8 diagnostics-13-00736-f008:**
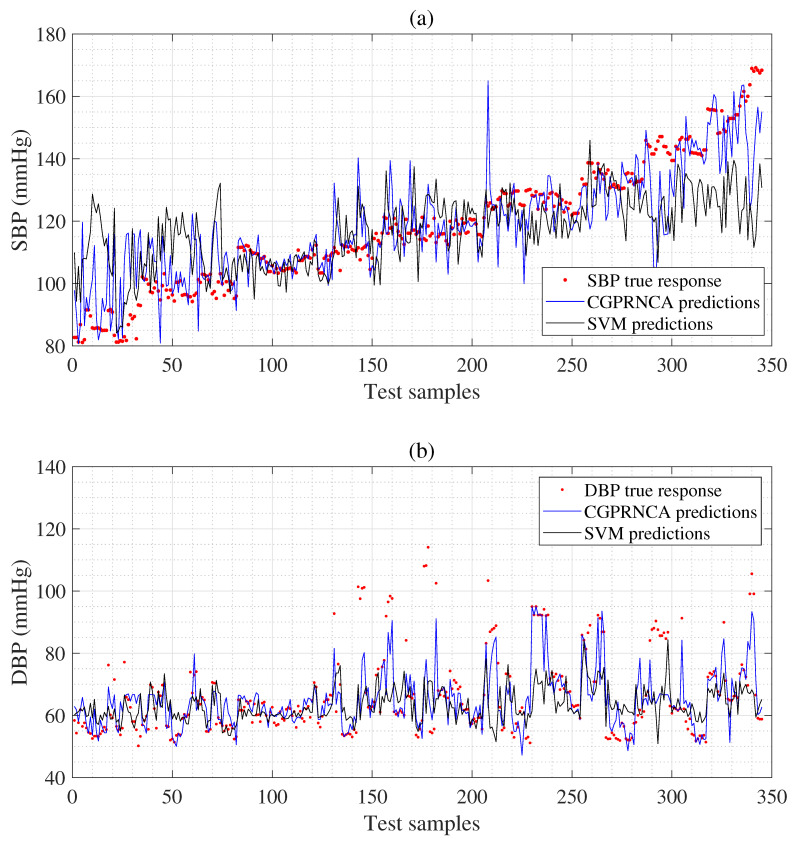
Top panel (**a**) compares the performance between the proposed CGPRNCA and SVM concerning the reference SBP; bottom panel (**b**) compares the performance between the proposed CGPRNCA and SVM for the reference DBP.

**Figure 9 diagnostics-13-00736-f009:**
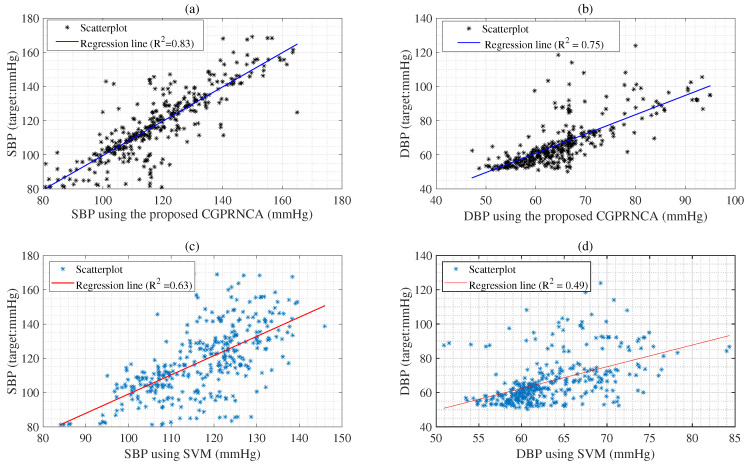
Top panel (**a**) shows the regression line between the proposed CGPRNCA and reference SBP (mmHg); panel (**b**) denotes the regression line between the proposed CGPRNCA and reference DBP (mmHg); panel (**c**) denotes the regression line between the SVM and reference SBP (mmHg); panel (**d**) denotes the regression line between the SVM and reference DBP (mmHg).

**Table 1 diagnostics-13-00736-t001:** Summary of the features (I).

Features	Explanation	Ref.
1: Systolic time (ST)	Ascending time from the trough of PPG to its systolic peak	[8,20]
2: Diastolic time (DT)	Descending time from PPG systolic peak to the next PPG	
	morphology diastolic trough	[8,20]
3: Pulse intensity ratio (PIR)	Ratio of the intensity of the PPG’s systolic peak	
	and diastolic trough	[8,20]
4: Heart rate (HR)	The inverse value of the duration between consecutive ECG’s R peaks	[8,20,39]
5: Pulse arrival time (PAT1)	The time between the R peak of ECG and the systolic peak of PPG	[20,39]
6: PAT(3)	The time between the R peak of ECG and diastolic trough	[20,39]
7: PAT(2)	The time between the R peak of ECG and	
	maximum slope point (1st derivative peak value)	[20,39]
8: Large artery stiffness index (LASI)	The inverse of the period from the PPG’s systolic peak to	
	the inflection point closest to the diastolic peak	[20,39]
9: Augmentation index (AI)	Measure of the pressure waves reflection on arteries and	
	it is computed through the ratio of the PPG pulse peak intensity and	
	the intensity of the inflection point closer to the diastolic peak	[20,39]

**Table 2 diagnostics-13-00736-t002:** Summary of the features (II), where $p_{m}$ denotes the area under divided by the pulse duration, $p_{d}$ is the minimum intensity of PPG, and $p_{s}$ denotes the maximum intensity of PPG.

Features	Explanation	Ref.
10: S1	Area under the PPG pulse curve from the diastolic trough to	
	the point of max slope	[20,39]
11: S2	From the point of max slope to the systolic peak	[20,39]
12: S3	From the systolic peak to the inflexion point closest to the diastolic peak	
13: S4	From the inflexion point to the next pulse’s diastolic trough	[39]
14: Inflection point area ratio (IPAR)	Ratio of S4/(S1 + S2 + S3)	[20,39]
15: ${\mathrm{PPG}}_{k}$	$\left(p_{m}-pd\right)/\left(p_{s}-pd\right)$	[20]
16: dPPG height (H)	PPG’s 1st derivative characteristics	[8,39]
17: dPPG width (W)	PPG’s 1st derivative characteristics	[8,20]
18: ddPPG peak height (PH)	PPG’s 2nd derivative characteristics	[8,20]
19: ddPPG trough height (TH)	PPG’s 2nd derivative characteristics	[8,20]
20: ddPPG width (W)	PPG’s 2nd derivative characteristics	[8,20]
21: ddPPG height (H)	PPG’s 2nd derivative characteristics	[8,20]
22: MXAP	The pulse’s maximum amplitude	[48]
23: MIAP	The pulse’s minimum amplitude	[48]
24: MEU	The blood’s viscosity	[48]
25: FHR	The frequency of HR	[48]

**Table 3 diagnostics-13-00736-t003:** The high score ranked features were selected using the F-test and RNCA algorithms for the SBP and DBP estimations.

	F-Test		RNCA			F-Test		RNCA	
Rank	SBP	DBP	SBP	DBP	Rank	SBP	DBP	SBP	DBP
1	dppgH	HR	HR	HR	14	S2	S4	PPGk	MXAP
2	ddppgPH	PAT1	ST	ST	15	PPGk	PIR	S4	IPA
3	PAT2	dppgH	DT	ddppgH	16	MXAP	MEU	FHR	MIAP
4	ddppgH	ddppgPH	PAT3	AI	17	DT	IPA	MXAP	MEU
5	PAT1	PAT3	ddppgW	PAT1	18	MEU	DT	MEU	DT
6	ddppgFH	ST	ddppgFH	PAT3	19	LASI	dppgW	IL	dppgH
7	ST	ddppgH	AI	S2	20	AI	FHR	IPA	ddppgPH
8	PAT3	LASI	PAT1	ddppgW	21	PIR	ddppgW	dppgW	S4
9	HR	PAT2	S3	ddppgFH	22	FHR	MXAP	PAT2	S1
10	S3	S3	S2	FHR	23	S4	S1	LASI	LASI
11	dppgW	ddppgFH	ddppgH	dppgW	24	IPA	MIAP	S1	PIR
12	ddppgW	S2	dppgH	S3	25	MIAP	AI	PIR	PAT2
13	S1	PPGk	ddppgPH	PPGk					

**Table 4 diagnostics-13-00736-t004:** Summarized parameters of the proposed and conventional algorithms, where SE is a squared exponential kernel for the GP algorithm.

Parameters	SVM	ANN	GP	GP	CGP	CGP	CGP
Hybrid				RNCA	F-Test	MRMR	RNCA
Number of samples	1725	1725	1725	1725	1725	1725	1725
Number of features	25	25	25	10–16	10–16	10–16	8–16
Output dimension	1	1	1	1	1	1	1
Optimizer	Bayes	Bayes	Bayes	Bayes	Bayes	Bayes	Bayes
Epsilon	1.00 × 10${}^{-08}$ − 0.01	-	-	-			
Fixed Weight Threshold	-	-	-	1 to 3	-	-	-
Shrinkage Factor	0.05−0.1	-	-	-	-	-	-
Subsampling Factor	0.10.5	-	-	-	-	-	-
Kernel Function	Gauss.	-	SE	SE	SE	SE	SE

**Table 5 diagnostics-13-00736-t005:** Compared feature training and testing times between the proposed and conventional methods based on H/W (Intel^®^ Core(TM) i5-9400 CPU 4.1 GHz, OS 64 bit, RAM 16.0 GB), and S/W (Matlab^®^ 2022 (The MathWorks Inc., Natick, MA, USA) specifications).

Algorithm	SVM	ANN	GP	GP	CGP	CGP	CGP
Hybrid			RNCA	F-Test	MRMR	RNCA	
Time (s)	0.58	4.50	3.72	40.52	287.35	282.14	292.35

**Table 6 diagnostics-13-00736-t006:** Reference ABP ranges in the dataset, where AAMI/ESH/ISO protocol defines the general population [46].

(mmHg)	Mean	STD	Min	Max	≥160	≥140	≤100	≥100	≥85	≤60
	(mmHg)	(mmHg)	(mmHg)	(mmHg)						
SBP	118.2	19.5	80.2	169.2	2.1%	15.4%	17.9%			
DBP	65.8	12.8	50.2	128.2				2.3%	10.6%	38.0%
AAMI/ESH/ISO					5%	20%	5%	5%	20%	5%

**Table 7 diagnostics-13-00736-t007:** ME and SDE relative to the reference ABP [46,50] and the conventional SVM, ANN, GP, and GP with RNCA, combining GP with F-test, GP with MRMR, and GP with RNCA algorithms, where CGP denotes combining Gaussian process.

Method	SVM		ANN		GP		GP		CGP		CGP		CGP	
							RNCA		F-Test		MRMR		RNCA	
(mmHg)	SBP	DBP	SBP	DBP	SBP	DBP	SBP	DBP	SBP	DBP	SBP	DBP	SBP	DBP
ME	0.03	−0.11	−0.04	−0.06	−0.04	0.63	0.06	−0.04	−0.02	−0.26	−0.03	−0.09	0.16	−0.03
SDE	13.33	11.50	13.04	10.32	11.80	9.61	10.85	8.42	11.62	9.44	11.31	8.89	10.79	8.07

**Table 8 diagnostics-13-00736-t008:** MAE (SDE) relative to the reference ABP and the conventional SVM, ANN, GP, GP with RNCA, combining GP with F-test, GP with MRMR, and GP with RNCA algorithms, where CGP denotes combining the Gaussian process, and mmHg denotes the unit measure for BP.

Method	SVM		ANN		GP		GP		CGP		CGP		CGP	
							RNCA		F-Test		MRMR		RNCA	
(mmHg)	SBP	DBP	SBP	DBP	SBP	DBP	SBP	DBP	SBP	DBP	SBP	DBP	SBP	DBP
MAE	9.84	8.30	9.75	7.23	8.39	6.45	7.38	5.50	7.83	5.97	7.85	5.82	7.22	5.18
SDE	2.63	1.34	2.97	0.39	0.43	0.34	0.49	0.39	0.64	0.43	0.41	0.36	0.44	0.39

**Table 9 diagnostics-13-00736-t009:** We use the results of SVM, ANN, GP, GP with RNCA, CGP with F-test, CGP with MRMR, and CGP with RNCA algorithms to grade the algorithm based on the BHS standard [3], where each result represents the average of 30 experimental data.

		SBP			DBP			SBP/DBP
Method	Hybrid	Mean Absolute Difference (%)			Mean Absolute Difference (%)			BHS
		≤5 mmHg	≤10 mmHg	≤15 mmHg	≤5 mmHg	≤10 mmHg	≤15 mmHg	Grade
SVM		39.48	63.59	77.31	42.36	70.73	87.53	-/C
ANN		36.55	62.64	78.94	51.06	76.62	88.03	-/C
GP		46.36	69.07	82.36	52.42	79.84	89.60	-/B
GP	RNCA	52.64	74.57	85.98	63.67	83.54	92.37	C/B
CGP	F-TEST	51.75	72.69	83.79	61.38	81.61	90.61	-/B
CGP	MRMR	49.52	71.97	84.58	60.82	83.01	91.39	-/B
CGP	RNCA	54.58	75.31	86.21	65.78	84.83	93.21	C/B
Grade A		60	85	95	60	85	95	[3]
Grade B		50	75	90	50	75	90	
Grade C		40	65	85	40	65	85	

**Table 10 diagnostics-13-00736-t010:** RMSE (SDE) and $R^{2}$ (SDE) relative to the reference ABP and the conventional SVM, ANN, GP, GP with RNCA, combining GP with F-test, GP with MRMR, and GP with RNCA algorithms, where CGP denotes combining Gaussian process, where (mmHg) is the unit measure for BP.

Method	SVM		ANN		GP		GP		CGP		CGP		CGP	
							RNCA		F-Test		MRMR		RNCA	
(mmHG)	SBP	DBP	SBP	DBP	SBP	DBP	SBP	DBP	SBP	DBP	SBP	DBP	SBP	DBP
RMSE	13.27	11.46	12.98	10.27	11.75	9.55	10.79	8.38	11.59	9.41	11.25	8.85	10.75	8.02
SDE	2.64	1.34	0.63	0.63	0.58	0.62	0.55	0.53	0.87	0.66	0.52	0.57	0.62	0.63
$R^{2}$	0.68	0.28	0.74	0.59	0.80	0.67	0.83	0.75	0.80	0.68	0.82	0.72	0.83	0.77
SDE	0.25	0.31	0.03	0.04	0.02	0.04	0.02	0.03	0.03	0.04	0.02	0.03	0.02	0.03

## Data Availability

Please refer to suggested Data Availability Statements in section “MDPI Research Data Policies” at https://archive.ics.uci.edu/ml/datasets/Cuff-Less+Blood+Pressure+Estimation (accessed on 1 June 2022). Upon a reasonable request, the corresponding author can offer a partial code for the study upon completion of all projects.

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
