# Peer review of "Combining Gaussian Process with Hybrid Optimal Feature Decision in Cuffless Blood Pressure Estimation"

_diagnostics, 2023, doi:10.3390/diagnostics13040736_

Round 1

Reviewer 1 Report

The paper discusses a methodology combining the Gaussian process with CGHOFD to estimate blood pressure. The paper is well organized and described.

Strengths: approach to combine methods, features extraction by analyzing ECG/PPG signal.

Points of weakness: explanation of the “pattern” followed to find the best combination of the approaches; algorithm performance discussion; state of the art; discussion of perspectives and limitations;

A minor revision is required.

Actions to do:

According to the weaknesses, I suggest to improve the paper by answering to these points:

1-     Please provide more information about the points that suggested you to find the final solution of combining approaches (testing patterns, preliminary test, etc.);  

2-     The ‘Discussion’ section should add more comments about the performance indicators:

-        in Table 7 the MAE is in %?,

-        explain/indicate the scale of the RMSE in table 9 (is it in %)

-         I suggest to add at least a  graph overlapping the MAE with the signal;

3-     More references should be added in the introduction section about an overview of AI image processing extracting features (AI applied on blood pressure signals), such as:

-        doi: 10.1109/MetroInd4.0IoT48571.2020.9138258

-        https://doi.org/10.3390/bios12040234

-        https://doi.org/10.1155/2021/5078799

4-     Explain better how you classify/predict the hypertension risk by the features extraction.

Minor remarks:

Please add more comment in all figure captions.

Author Response

Response to the first reviewers’ comments

“Combining Gaussian Process with Hybrid Optimal Feature Decision in Cuff-less Blood Pressure Estimation.”

Soojeong Lee, Gyanendra Prasad Joshi, Chang-Hwan~Son, Gangseong Lee,

Department of Computer Engineering, Sejong University, Seoul

Manuscript ID: diagnostics-2136893

General

We appreciate the valuable comments and suggestions of the reviewers on our paper very much. We have incorporated all the reviewers’ comments and suggestions in our submitted manuscript and given additional explanations. Our detailed responses are as follows.

  1. According to the comments, “Please provide more information about the points that suggested you to find the final solution of combining approaches (testing patterns, preliminary test, etc.);” 

Answer 1, we provided and revised more information about the final solution of combining approaches as

“As mentioned in the section of RNCA, the best regularization parameter is obtained using steps 05 to 10 in Algorithm 3. The weighted feature vectors are then prepared using the RNCA, as shown in steps 11 to 12 of Algorithm 3.

Afterward, we initialize variables to call the HOFD function shown in steps 13 to 16. Specifically, the weighted features were descending and sorted as in step 13.

Next, the HOFD is called to compute the least root mean square error (RMSE) in step 17, where $X$ denotes the training features $y$ is the reference of SBP. As shown in steps 18 to 20, we can obtain the least RMSE and the number of weighted features. To find the least RMSE, we continue to call the HOFD function by decreasing the number of weighted features by one in steps 17 to 21. Finally, the optimal feature index is determined to find the least RMSE in steps 22 to 25.

The proposed HOFD function starts from a set of weighted features obtained from RNCA and sequentially includes each feature that still needs to be selected to create a subset of candidate features. The HOFD conducts 5-fold cross-validation by repeatedly calling GP with different training subsets of X and y, as in step 03 of Algorithm 4. Here, Xtrain and ytrain contain identical X and y, while Xtest and ytest include the complementary subset of rows. Xtrain and Xtest have the features acquired from the columns of X that correspond to the current candidate feature set as in steps 04 to 07. Afterward,

each time it is called, the GP should return a model criterion as in steps 08 to 09. 

Typically, GP uses Xtrain and ytrain to train, then predicts values for Xtest using that GP model in steps 10, and finally returns some measure of loss RMSE of those predicted values from ytest in steps 11 to 12. After computing the mean criterion values for each candidate feature subset, HOFD chooses the candidate feature subset that minimizes the RMSE criterion value in step 13 of Algorithm 4, which is used to determine the best feature subset as given in (Algorithm 3).”

Hence, combining GP with HOFD leads to an effective feature selection process. In the first process of our algorithm, the filter-based algorithm is applied to find a weighted feature set. In the second process, GP directly determines the best feature subset from the weighted feature set. Therefore, we obtain the best feature subset with high weights from the original feature set as given in (Algorithms 3 and 4).”

Lines 206-234.

  1. According to the comments, “The ‘Discussion’ section should add more comments about the performance indicators:

-        in Table 7 the MAE is in %?,

-        explain/indicate the scale of the RMSE in table 9 (is it in %)

-         I suggest to add at least a  graph overlapping the MAE with the signal;”

    Answer 2, we revised it in the discussion as

“Table 8. MAE (SDE) relative to the reference ABP and the conventional SVM, GP, GP with RNCA, combining GP with F-test, GP with MRMR, and GP with RNCA algorithms, where CGP denotes combining Gaussian process, and (mmHg) denotes the unit measure for BP.”

“Table 10. RMSE (SDE) and R^{2} (SDE) relative to the reference ABP and the conventional SVM, ANN, GP, GP with RNCA, combining GP with F-test, GP with MRMR, and GP with RNCA algorithms, where CGP denotes combining Gaussian process, where (mmHg) is the unit measure for BP.”

We included a graph that compared the proposed method with the conventional SVM, as shown in Figure 8.

“Regarding estimation accuracy, the proposed CGPRNCA algorithm exhibits the lowest MAE for SBP (7.22 mmHg) and DBP (5.18 mmHg) compared with the reference BPs given in Table 8. In particular, SBP 2.2%= (7.38 - 7.22) / 7.22 * 100  and DBP 6.2%= (5.50 - 5.18) / 5.53 * 100  were improved compared to the MAEs of SBP (7.38 mmHg) and DBP (5.50 mmHg) in the GPRNCA algorithm, proving that the proposed CGPRNCA algorithm is effective in estimating SBP and DBP. We confirm the effect of the HOFD process that finally determines the weighted feature subset.

In addition, when the proposed CGPRNCA algorithm is compared with the GP algorithm, it shows an improved performance of 16.2%= (8.39  - 7.22) / 7.22 * 100 for SBP and 24.5%= (6.45 - 5.18) / 5.18 * 100 for DBP. ~~~”

Lines 364-376.

  1. According to the comments “ 3-More references should be added in the introduction section about an overview of AI image processing extracting features (AI applied on blood pressure signals), such as:

-        doi: 10.1109/MetroInd4.0IoT48571.2020.9138258

-        https://doi.org/10.3390/bios12040234

-        https://doi.org/10.1155/2021/5078799”

Answer 3, we added the references in the introduction section as

“Massaro et al. [10] proposed a decision support system for estimating health status based on artificial intelligence algorithms.  A novel smart healthcare monitoring system using ML and the internet of things was developed by [11], which created an automated artifact detection method for BP and PPG signals. Tan et al. [12] introduced an artificial intelligence-enhanced BP monitoring wristband. The wristband’s sensors are based on piezoelectric nanogenerators.”

Lines 32-37, pages 1-2.

  1. According to the comments, “4-Explain better how you classify/predict the hypertension risk by the features extraction.”

Answer 4, we included it as “In addition, the proposed methodology can constantly monitor BP change through the continuous variability of RMSE's SDEs and MAE's SDEs to estimate hypertension risk. The proposed HOFD process based on the GP algorithm may effectively be used for BP estimation.”

Lines 434-437.

Reviewer 2 Report

Most of the patients consider cuff-dependent blood pressure monitoring obtrusive and boarding, especially at night and from long-term monitoring perspective, that reduce broader utilization and quality of the existing blood pressure monitoring devices. The problem of developing solutions for cuffless blood pressure estimation and validation of devices based on such techniques thus makes up a vital and topical agenda.

This paper suggests a novel approach to help solve this problem.

Though the method itself can be promising, there are several methodical issues that need consideration before I recommend paper for publication.

1. Off-scope left key papers on the cuffless blood pressure estimation and evaluation guidelines to assess data accuracy and validness, e.g., doi:10.1097/HJH.0000000000003224; doi:10.1161/HYPERTENSIONAHA.121.17747; doi: 10.3389/fcvm.2019.00040; doi: 10.1093/ajh/hpac017.). Albeit recognizing that technologies underlying cuffless blood pressure devices have considerable potential to improve the awareness, treatment, and management of hypertension, recent guidelines by the European Society for Hypertension (ESH) do not recommend cuffless devices for the diagnosis and management of hypertension (see also, US (DOI: 10.1093/ajh/hpac017) and Korean (doi: 10.1186/s40885-020-00158-8) official position). Three major challenges in validation of cuffless blood pressure measurement devices (doi:10.1161/HYPERTENSIONAHA.121.17747): adherence to established validation protocols (doi: 10.1097/HJH.0000000000001634.), the inclusion of inter- and intra-individual blood pressure variations. These obstacles are recommended to be addressed in discussion.

2. Why authors made a decision to use public data sets which only allowed “the relatively small sample size for each patient”, as stated at P.22, L.400, and thus restricted opportunity to test for inter- and intra-individual blood pressure variations?

3. Have authors compared estimation of values obtained at a different time at 24-h (circadian) scale, since it is known that all cardiovascular functions vary in mean and deviations across these scales?

4. On page 3, L.131 the authors stated that “only used the first 20 seconds (s) of the record because each record in the database ranges from a minimum of 20 (s) to 530” and on L.130 “...omitted data from specific subjects with very high and very low BPs as follows: (SBP >= 180, SBP <= 80, DBP >= 130, and DBP <= 50)”. Each of these decisions is at least questionable for validation of cuffless blood pressure examination on two reasons: 1. first measurements obtained from the patient are usually the least reliable and predictable and recommended to be omitted in algorithms of home blood pressure analyses (e.g., DOI: 10.1097/01.hjh.0000239289.87141.b6), 2. excluded range of BP values are common for ambulatory real-life conditions (high, related to physical activity and in advanced stages hypertension patients; low, nocturnal values and sleep). These values are quite important for any BP estimation purposes that supposed to include 24-hour ambulatory observations.

Author Response

Response to the second reviewers’ comments

“Combining Gaussian Process with Hybrid Optimal Feature Decision in Cuff-less Blood Pressure Estimation.”

Soojeong Lee, Gyanendra Prasad Joshi, Chang-Hwan~Son, Gangseong Lee,

Department of Computer Engineering, Sejong University, Seoul

Manuscript ID: diagnostics-2136893

  1. According to the comments,

Off-scope left key papers on the cuffless blood pressure estimation and evaluation guidelines to assess data accuracy and validness, e.g., doi:10.1097/HJH.0000000000003224; doi:10.1161/HYPERTENSIONAHA.121.17747; doi: 10.3389/fcvm.2019.00040; doi: 10.1093/ajh/hpac017.).

Albeit recognizing that technologies underlying cuffless blood pressure devices have considerable potential to improve the awareness, treatment, and management of hypertension, recent guidelines by the European Society for Hypertension (ESH) do not recommend cuffless devices for the diagnosis and management of hypertension (see also, US (DOI: 10.1093/ajh/hpac017) and Korean (doi: 10.1186/s40885-020-00158-8) official position).

Three major challenges in validation of cuffless blood pressure measurement devices (doi:10.1161/HYPERTENSIONAHA.121.17747): adherence to established validation protocols (doi: 10.1097/HJH.0000000000001634.), the inclusion of inter- and intra-individual blood pressure variations. These obstacles are recommended to be addressed in discussion.

Answer 1, we experimented again and included more information about the evaluation guidelines to validate protocols as

“Table 6. Reference ABP ranges in the data set, where AAMI/ESH/ISO protocol defines the general population [42].”

Page 16.

“Table 7. ME and SDE relative to the reference ABP [42], [44]and the conventional SVM, ANN, GP, GP with RNCA, combining GP with F-test, GP with MRMR, and GP with RNCA algorithms, where CGP denotes combining Gaussian process.”

Page 17.

“Table 8. MAE (SDE) relative to the reference ABP and the conventional SVM, ANN, GP, GP with RNCA, combining GP with F-test, GP with MRMR, and GP with RNCA algorithms, where CGP denotes combining Gaussian process, and (mmHg) denotes the unit measure for BP.”

Page 18.

“Table 9. We use the results of SVM, ANN, GP, GP with RNCA, CGP with F-test, CGP with MRMR, and CGP with RNCA algorithms to grade the algorithm based on the BHS standard [3], where each result represents the average of 30 experimental data.”

Page 19.

“Table 10. RMSE (SDE) and R^2 (SDE) relative to the reference ABP and the conventional SVM, ANN, GP, GP with RNCA, combining GP with F-test, GP with MRMR, and GP with RNCA algorithms, where CGP denotes combining Gaussian process, where (mmHg) is the unit measure for BP.”

Answer 2. We included additional figures 5, 7, 8, and 9 to validate protocols as

“Figure 5. Histogram of the database, where (a) denotes the reference SBP (mmHg) and (b) denotes the reference DBP (mmHg).”

“Figure 7. Top panel (a) shows comparing performance between the proposed CGPRNCA and reference SBP (mmHg); panel (b) denotes comparing performance between the proposed CGPRNCA and reference DBP (mmHg); panel (c) denotes comparing performance between the SVM and reference SBP (mmHg); panel (d) shows comparing performance between the SVM and reference DBP (mmHg).”

“In particular, SBP 2.2%= (7.38  - 7.22) / 7.22 * 100 and DBP 6.2= (5.50 - 5.18) / 5.53 * 100 were improved compared to the MAEs of SBP (7.38 mmHg) and DBP (5.50 mmHg) in the GPRNCA algorithm, proving that the proposed CGPRNCA algorithm is effective in estimating SBP and DBP. We confirm the effect of the HOFD process that finally determines the weighted feature subset. In addition, when the proposed CGPRNCA algorithm is compared with the GP algorithm, it shows an improved performance of 16.2%= (8.39  - 7.22) / 7.22 * 100 for SBP and 24.5%= (6.45 - 5.18) / 5.18 * 100 for DBP. On the other hand, the SDEs of MAEs in all algorithms show stable values except for the SVM, as given in Table \ref{tab8}. The comparison between the SBP estimation result of the proposed CGPRNCA algorithm and the estimation result of the conventional SVM is shown in Figure 8.”

Lines 366-376.

We experimented again as

“Figure 6. We compared the performance between the SVM, ANN, and GP with respect to the reference ABP for the SBP (a) and DBP (c); We also compared the performance between the GPRNCA and CGP with F-test concerning the reference ABP for the SBP (b) and DBP (d);  We compared the performance between the CGP with MRMR and CGP with RNCA with respect to the reference ABP for the SBP (b) and DBP (d); Finally, compared the performance between the GP with RNCA and CGP with RNCA with respect to the reference ABP for the SBP (b) and DBP (d).”

We included the limitation of our research to validate protocols as

“Although we experimented with a public MIMIC-II dataset, this study showed limitations in that patient distribution, and experimental results did not satisfy the AAMI/ESH/ISO protocols [43].

Based on the AAMI/ESH/ISO protocol, we evaluated the validity of the proposed method.

Although the MIMIC II satisfies the standard of 85 subjects in general, the MIMIC II data set does not meet the AAMI/ESH/ISO protocol at SBP >160 mmHg 5%, >140 mmHg 20%, and DBP >100 mmHg 5\%, >85 mmHg 20% [43]. The high blood pressure part shows a small distribution, and the low blood pressure part shows a large distribution as 17.9% for SBP 100 mmHg or less and 38.0% for DBP 60 mmHg or less. Moreover, obtaining inter- and intra-individual BP variations is crucial for cuffless device evaluation but challenging to acquire [51]. Finally, the subject population should exhibit a wide BP range for calibration-free algorithms, such as those required by universal standards. However, identifying these populations can be difficult and costly, and meeting many of these criteria will require much work, especially in lab-scale studies of popular cuffless devices [51]. We believed that MIMIC II data were obtained from long-term monitoring of patients. However, further research should be considered when obtaining data based on the AAMI/ESH/ISO protocol.

Therefore, our laboratory will conduct a confidence interval of BP estimation to track the inter-and intra-individual BP variations to observe the cardiovascular functions vary at 24 hours. In addition, we should obtain more BP data and conduct experiments to verify intra-individual blood pressure changes according to the  AAMI/ESH/ISO protocol.

Moreover, since calibration-free algorithms often use age and gender as inputs to ML models, this causes the accuracy of the experimental results to be unclear [51]. However, in this study, age and gender were not used as input values for the proposed algorithm.

Nevertheless, the following study should improve the performance of the CGPRNCA algorithm by reducing the SDE of ME to pass the AAMI criteria [45]. Another disadvantage is that the complexity of the HOFD algorithm should be improved for BPs estimation.”

  1. According to the comments, “Why authors made a decision to use public data sets which only allowed “the relatively small sample size for each patient”, as stated at P.22, L.400, and thus restricted opportunity to test for inter- and intra-individual blood pressure variations?”

Answer 2. We agreed with the comment thus, we removed the sentence and answered as 

“Therefore, our laboratory will conduct a blood pressure confidence interval estimation to track the inter-and intra-individual BP variations to observe the cardiovascular functions vary at 24 hours. We should obtain more BP data and conduct experiments to verify intra-individual blood pressure changes according to the AAMI/ESH/ISO protocol.”

Lines 454-457.

  1. According to the comments, “3. Have authors compared estimation of values obtained at a different time at 24-h (circadian) scale, since it is known that all cardiovascular functions vary in mean and deviations across these scales?”

Answer 3. We agreed with the comments, but we believed that MIMIC II data was obtained from long-term monitoring of patients, and we included as

“We believed that MIMIC II data were obtained from long-term monitoring of patients. However, further research should be considered when obtaining data based on the AAMI/ESH/ISO protocol.

Therefore, our laboratory will conduct a confidence interval of BP estimation to track the inter-and intra-individual BP variations to observe the cardiovascular functions vary at 24 hours. In addition, we should obtain more BP data and conduct experiments to verify intra-individual blood pressure changes according to the AAMI/ESH/ISO protocol [42].”

Lines 451-457.

  1. According to the comments “On page 3, L.131 the authors stated that “only used the first 20 seconds (s) of the record because each record in the database ranges from a minimum of 20 (s) to 530” and on L.130 “...omitted data from specific subjects with very high and very low BPs as follows: (SBP >= 180, SBP <= 80, DBP >= 130, and DBP <= 50)”. Each of these decisions is at least questionable for validation of cuffless blood pressure examination on two reasons: 1. first measurements obtained from the patient are usually the least reliable and predictable and recommended to be omitted in algorithms of home blood pressure analyses (e.g., DOI: 10.1097/01.hjh.0000239289.87141.b6), 2. excluded range of BP values are common for ambulatory real-life conditions (high, related to physical activity and in advanced stages hypertension patients; low, nocturnal values and sleep). These values are quite important for any BP estimation purposes that supposed to include 24-hour ambulatory observations.”

Answer 4. We obtained the data from the PPG, ECG, and ABP (mmHg) again and revised the sentence as

“Because each record range in the database was different, each record after 60 (s) was used to increase the reliability of the records obtained from the patients [42].  We omitted records from specific subjects with very high and low BPs to remove abnormal outlier records, such as moving artifacts as follows: (SBP >=180, SBP <=80, DBP>=130, and DBP <=50).”

Lines 119-129.

Reviewer 3 Report

Dear Authors,

Congratulations on your paper, which is about an interesting issue. However, there are some improvements that should be done before it is accept for publication. Thus, pleas econsider my following comments and suggestions:

1. Please refer to the gap found in the literature which has motivate this work.

2. Please state clearly the novelty of your work regarding previous developments in this field of knowledge.

3. Readers usually prefer direct speech in the literature review, as you are doing with reference [10]. Please extend this type of speech to other works worth of mention, instead generic ideas supported by a large number of references.

4. In Page 1/Line 34, after the name of the Authot ( Z. Qiu), please insert the reference ([8]), not leaving to the end of the sentence. Moreover, please use just the last name of the first Author.

5. (Page 2 - Top) Please state the reasons behind the use of the random forest based regression network and the three-layer ANN-based regression network, and the SVM model. Why not others?

6. In my opinion, your contributions shoul be pointed out at the end of the paper (Conclusions). Please consider reorganize your paper in this way.

7. The methodology used is very well described. Congratulations. However, the data is not contextualized in terms of sampling. Where the data were taken? Range of old? And so on. Could you complete? I think this information is important, not to support the model development, but to understand the type of data considered.

8. Training data is a usual problem regarding ANN accuracy results. Did you perform any computing experience with different training data?

9. Please try to present the statistical results in a systematic way using tables, leaving the explanations to the Discussion.

10. Discussion is very well carried out, but could be improved (see previous point).

11. No real values are pointed out in the Conclusions regarding the accuracy of the model and main results. Please improve.

Best wishes,

FGS

Author Response

Response to the third reviewers’ comments

“Combining Gaussian Process with Hybrid Optimal Feature Decision in Cuff-less Blood Pressure Estimation.”

Soojeong Lee, Gyanendra Prasad Joshi, Chang-Hwan~Son, Gangseong Lee,

Department of Computer Engineering, Sejong University, Seoul

Manuscript ID: diagnostics-2136893

  1. According to the comments, “Please refer to the gap found in the literature which has motivate this work.”

Answer 1. We included it as

“Massaro et al. [10] proposed a decision support system for estimating health status based on artificial intelligence algorithms.  A novel smart healthcare monitoring system using ML and the internet of things was developed by [11], which created an automated artifact detection method for BP and PPG signals.  Tan et al. [12] introduced an artificial intelligence-enhanced BP monitoring wristband. The wristband’s sensors are based on piezoelectric nanogenerators.”

Lines 32-37, pages 1-2.

“The critical issue to increasing ML algorithms' reliability is extracting features essential to the response variable [20-22].”

Lines 46-47, page 2.

“Another issue for improving the performance of ML algorithms is feature selection to use as input data by replacing the original features [23-24, 26-28].”

Lines 54-55, page 2.

  1. According to the comments, “Please state clearly the novelty of your work regarding previous developments in this field of knowledge.”

Answer 2. We revised the novelty of our work as

“This study contributes to BPs estimation as follows. First, the proposed combining Gaussian process with a hybrid optimal feature decision (CGHOFD) algorithm is developed to improve reliability in cuff-less BP estimation.  Second, the proposed methodology using the hybrid optimal feature decision (HOFD) process overcomes the limitation of missing valuable features, a limitation of the conventional filter-based feature selection algorithm.

Third, the adaptive HOFD algorithm automatically uses GP evaluation criteria to decide the best feature subset. Fourth, the proposed method solves the problem of specifying the number of weights when selecting weighted features and the RNCA problem using a fixed threshold.”

Lines 474-782.

  1. According to the comments, “Readers usually prefer direct speech in the literature review, as you are doing with reference [10]. Please extend this type of speech to other works worth of mention, instead generic ideas supported by a large number of references.”

Answer 3. We included the literature review as

“Massaro et al. [10] proposed a decision support system for estimating health status based on artificial intelligence algorithms.  A novel smart healthcare monitoring system using ML and the internet of things was developed by [11], which created an automated artifact detection method for BP and PPG signals. Tan et al. [12] introduced an artificial intelligence-enhanced BP monitoring wristband. The wristband’s sensors are based on piezoelectric nanogenerators.”

Lines 32-37, pages 1-2.

“Qiu et al. [8] proposed a new method for estimating BP using a window function-based piecewise neural network. This paper evaluated the random forest-based regression network and the three-layer ANN-based regression network, and the SVM model as a less complex algorithm using the PPG signal in order to perform the accurate cuffless BP estimation. Here, valuable features were extracted using the PPG signal's first and second derivative waveforms and used as input data for BP estimation.”

Lines 40-46, page 2.

  1. According to the comments, “In Page 1/Line 34, after the name of the Authot ( Z. Qiu), please insert the reference ([8]), not leaving to the end of the sentence. Moreover, please use just the last name of the first Author.

Answer 4. We fixed it as

 “Qiu et al. [8] proposed a new method for estimating BP using a window function-based piecewise neural network.”

 Lines 41-42, page 2.

  1. According to the comments, “(Page 2 - Top) Please state the reasons behind the use of the random forest based regression network and the three-layer ANN-based regression network, and the SVM model. Why not others?”

Answer 5. We included it as

“This paper evaluated the random forest-based regression network and the three-layer ANN-based regression network, and the SVM model as a less complex algorithm using the PPG signal in order to perform the accurate cuffless BP estimation. Here, valuable features were extracted using the PPG signal's first and second derivative waveforms and used as input data for BP estimation.”

Lines 43-46, page 2.

  1. According to the comments, “In my opinion, your contributions should be pointed out at the end of the paper (Conclusions). Please consider reorganize your paper in this way.”

Answer 6. We reorganized our contributions as

“In conclusion, the proposed CGPRNCA improved accuracy and stability by using the HOFD process that reduces uncertainties such as MAE, SDE of mean error (ME), and RMSE for cuff-less SBP and DBP estimation. The proposed CGHOFD algorithm selects a filter suitable for feature characteristics using a hybrid mode to improve the disadvantages of the conventional filter-based method. Then, it is a combining method in which the selected weighted subsets are finally determined by finding the best feature subsets using the GP algorithm evaluation criterion. We performed extensive experiments to compare the conventional algorithms with the proposed CGHOFD algorithm on the public dataset for cuff-less BPs estimation. The experimental results confirm that the proposed CGHOFD algorithm is very effective.  This study contributes to BPs estimation as follows. First, the proposed combining Gaussian process with a hybrid optimal feature decision (CGHOFD) algorithm is developed to improve reliability in cuff-less BP estimation.  Second, the proposed methodology using the hybrid optimal feature decision (HOFD) process overcomes the limitation of missing valuable features, a limitation of the conventional filter-based feature selection algorithm. Third, the adaptive HOFD algorithm automatically uses GP evaluation criteria to decide the best feature subset. Fourth, the proposed method solves the problem of specifying the number of weights when selecting weighted features and the RNCA problem using a fixed threshold.”

  1. According to the comments, “The methodology used is very well described. Congratulations. However, the data is not contextualized in terms of sampling. Where the data were taken? Range of old? And so on. Could you complete? I think this information is important, not to support the model development, but to understand the type of data considered.”

Answer 7. We included it as

“We collected from the university of california irvine (UCI) ML repository center [40], which was extracted from MIMIC-II (Multiple Parameter Intelligent Monitoring) data [40-41]. The database consists of ECG, finger PPG, and ABP (arterial blood pressure) signals from 3000 records (subjects) at 125 Hz (sampling frequency). Reference systolic blood pressure (SBP) and diastolic blood pressure (DBP) were calculated from ABP signals, and the feature set was obtained by combining PPG with ECG signal waveforms.  Because each record range in the database was different, each record after 60 (s) was used to increase the reliability of the records obtained from the patients [42].  We omitted records from specific subjects with very high and low BPs to remove abnormal outlier records, such as moving artifacts as follows: (SBP >=180, SBP <=80, DBP>=130, and DBP <=50).”

Lines 120-129.

“In this paper, the MIMIC II database was used to extract the PPG, ECG, and ABP signals [40-41]. The first data set comprised 3000 records (subjects) with the PPG, ECG, and ABP signals. Because each record range in the database was different, each record after 60 (s) was used to increase the reliability of the records obtained from the patients [42].  After preprocessing, the final database was composed 1723 records with unique subjects, which satisfied the general population 85 subjects [43]. Table 6 and Figure 5  represented some statistical information about the range and distribution of the reference SBP and DBP in the final data set.”

Lines 264-271.

  1. According to the comments, “Training data is a usual problem regarding ANN accuracy results. Did you perform any computing experience with different training data?”

Answer 8. Yes, from 2004 to 2023, I have experience dealing with many biomedical data in AI and machine learning.

We included the results of the ANN algorithm as SDE of ME, MAE, RMSE …

  1. According to the comments “Please try to present the statistical results in a systematic way using tables, leaving the explanations to the Discussion.

We reorganized in discussion as

“This is the first study to propose combining the Gaussian process with hybrid optimal feature decision (HOFD) in cuff-less blood pressure estimation.  Table \ref{tab3} shows the high-scoring features selected using the F-test and RNCA algorithms  \cite{matlab2}, \cite{zhuang}. Here, we can see that the ranked features rely on the feature selection algorithm. Also, the ranked features were changed depending on the SBP or DBP target variable. Hence, we confirm the best feature decision using the proposed HOFD algorithm. Therefore, the proposed HOFD algorithm was more conducive to improving the performance of ML than using the threshold of the conventional RNCA algorithm. ~~~~”

Lines 313-436

  1. Discussion is very well carried out, but could be improved (see previous point).

We reorganized in discussion as

“This is the first study to propose combining the Gaussian process with hybrid optimal feature decision (HOFD) in cuff-less blood pressure estimation.  Table \ref{tab3} shows the high-scoring features selected using the F-test and RNCA algorithms  \cite{matlab2}, \cite{zhuang}. Here, we can see that the ranked features rely on the feature selection algorithm. Also, the ranked features were changed depending on the SBP or DBP target variable. Hence, we confirm the best feature decision using the proposed HOFD algorithm. Therefore, the proposed HOFD algorithm was more conducive to improving the performance of ML than using the threshold of the conventional RNCA algorithm. ~~~~”

Lines 313-436

  1. No real values are pointed out in the Conclusions regarding the accuracy of the model and main results. Please improve.

We revised the Conclusions as

“In conclusion, the proposed CGPRNCA improved accuracy and stability by using the HOFD process that reduces uncertainties such as MAE, SDE of mean error (ME), and RMSE for cuff-less SBP and DBP estimation. The proposed CGHOFD algorithm selects a filter suitable for feature characteristics using a hybrid mode to improve the disadvantages of the conventional filter-based method. Then, it is a combining method in which the selected weighted subsets are finally determined by finding the best feature subsets using the GP algorithm evaluation criterion. We performed extensive experiments to compare the conventional algorithms with the proposed CGHOFD algorithm on the public dataset for cuff-less BPs estimation. The experimental results confirm that the proposed CGHOFD algorithm is very effective.  This study contributes to BPs estimation as follows. First, the proposed combining Gaussian process with a hybrid optimal feature decision (CGHOFD) algorithm is developed to improve reliability in cuff-less BP estimation.  Second, the proposed methodology using the hybrid optimal feature decision (HOFD) process overcomes the limitation of missing valuable features, a limitation of the conventional filter-based feature selection algorithm. Third, the adaptive HOFD algorithm automatically uses GP evaluation criteria to decide the best feature subset. Fourth, the proposed method solves the problem of specifying the number of weights when selecting weighted features and the RNCA problem using a fixed threshold.”